# Flow Perturbation++: Multi-Step Unbiased Jacobian Estimation for High-Dimensional Boltzmann Sampling

## Abstract

The scalability of continuous normalizing flows (CNFs) for unbiased Boltzmann sampling remains limited in high-dimensional systems due to the cost of Jacobian-determinant evaluation, which requires $D$ backpropagation passes through the flow layers. Existing stochastic Jacobian estimators such as the Hutchinson trace estimator reduce computation but introduce bias, while the recently proposed Flow Perturbation method is unbiased yet suffers from high variance. We present **Flow Perturbation++**, a variance-reduced extension of Flow Perturbation that discretizes the probability-flow ODE and performs unbiased stepwise Jacobian estimation at each integration step. This multi-step construction retains the unbiasedness of Flow Perturbation while achieves substantially lower estimator variance. Integrated into a Sequential Monte Carlo framework, Flow Perturbation++ achieves significantly improved equilibrium sampling on a 1000D Gaussian Mixture Model and the all-atom Chignolin protein compared with Hutchinson-based and single-step Flow Perturbation baselines.

## 1. Introduction

Sampling from the Boltzmann distribution $p(\mathbf{x}) \propto e^{-u(\mathbf{x})}$ is a long-standing challenge in statistical mechanics and molecular modeling. Rugged, high-dimensional energy landscapes with numerous metastable states are separated by large free-energy barriers (Frauenfelder et al., 1991), causing classical Molecular Dynamics (MD) (Verlet, 1967) and Markov Chain Monte Carlo (MCMC) (Metropolis et al., 1953) methods to mix extremely slowly. Enhanced sampling techniques—such as replica exchange (Hukushima & Nemoto, 1996; Swendsen & Wang, 1986), umbrella sampling (Torrie & Valleau, 1977), metadynamics (Laio & Parrinello, 2002), and transition-path sampling (Dellago et al., 1998)—can mitigate these difficulties but often introduce substantial computational overhead or require carefully engineered, system-specific collective variables.

Beyond physical systems, Boltzmann distributions provide a unifying modeling principle in modern machine learning and reinforcement learning (RL) (Messaoud et al., 2024; Haarnoja et al., 2018; Chao et al., 2024). In RL, Boltzmann structures arise under entropy-regularized and inference-based formulations. In particular, maximum-entropy RL yields stochastic policies of the form $\pi(a \mid s) \propto \exp\big(Q^\pi(s,a)/\alpha\big)$, where action selection is governed by a soft action-value function rather than the immediate reward (Haarnoja et al., 2017; Shi et al., 2019; Jain et al., 2025). At the trajectory level, energy-based RL and control-as-inference methods define distributions $\pi(\mathbf{x}) \propto \exp\big(\beta R(\mathbf{x})\big)$, which underpin preference-based RL, reward-model fine-tuning, and trajectory generation (Bengio et al., 2023; Malkin et al., 2022; Zhu et al., 2025). Combined with their foundational role in statistical physics, this broad applicability motivates the development of efficient and scalable sampling methods.

This widespread demand has driven the development of deep generative models for learning Boltzmann target distributions and enabling efficient sampling. By transporting a simple base distribution to the target, such models have achieved notable success in molecular and material design, including drug-like molecule generation, protein conformations, and novel materials (Zheng et al., 2024; Abramson et al., 2024; Xu et al., 2022; Hoogeboom et al., 2022; Wang et al., 2024b; Noé et al., 2019).

Normalizing flows (NFs) (Rezende & Mohamed, 2015; Dinh et al., 2015; 2017; Kingma & Dhariwal, 2018) provide exact likelihood evaluation through invertible transformations between a simple prior and a complex target distribution. Discrete NFs rely on carefully designed architectures to make Jacobian determinants tractable, trading expressive flexibility for computational efficiency.

Continuous normalizing flows (CNFs) (Köhler et al., 2020), a continuous-time limit of NFs, parameterize transforma-

[1]Anonymous Institution, Anonymous City, Anonymous Region, Anonymous Country. Correspondence to: Anonymous Author <anon.email@domain.com>.

Preliminary work. Under review by the International Conference on Machine Learning (ICML). Do not distribute.

tions via neural ordinary differential equations (neural ODEs) (Chen et al., 2018) and have recently been applied to Boltzmann sampling (Noé et al., 2019; Klein et al., 2023; Wang et al., 2024a). A key advantage of CNFs is their ease of training, enabled by simulation-free objectives such as diffusion-based probability-flow ODEs (Ho et al., 2020; Song et al., 2021) and flow matching (Lipman et al., 2023), which have achieved strong empirical performance in image generation and molecular modeling.

Despite these advantages, unbiased Boltzmann sampling with CNFs remains challenging. To obtain unbiased samples from the Boltzmann target, one must correct the flow-induced proposal distribution via importance reweighting, $w(\mathbf{x}) = p(\mathbf{x})/q(\mathbf{x})$, which requires evaluating exact likelihood ratios. Evaluating importance weights requires integrating the trace of the Jacobian of the velocity field along the flow trajectory, necessitating $D$ backpropagation passes through all layers of the flow, where $D$ is the dimensionality of the system (Noé et al., 2019). While stochastic trace estimators such as the Hutchinson estimator (Hutchinson, 1989) reduce this cost (Wang et al., 2024a), they often introduce convergence issues and systematic bias (Noé et al., 2019; Erives et al., 2024). Consequently, existing CNF-based methods have only achieved unbiased Boltzmann sampling for relatively low-dimensional systems (Klein et al., 2023).

Building on Flow Perturbation (FP) (Peng & Gao, 2025), which estimates flow Jacobians via stochastic perturbations to reduce the cost of unbiased Boltzmann sampling, we introduce **Flow Perturbation++ (FP++)**, a theoretically grounded, unbiased, and variance-reduced estimator for diffusion-model dynamics. FP++ exploits the semigroup property of flow maps to decompose the global Jacobian $\det J_{1 \to T} = \prod_{k=1}^{T} \det J_k$, estimating each factor without computing the full Jacobian directly.

Our main contributions are:

- **Unbiased entropy estimation.** FP++ retains the unbiasedness of FP (Peng & Gao, 2025) while improving numerical stability.

- **Variance reduction.** Decomposing the global Jacobian into per-step determinants reduces variance by removing correlations between steps, especially for flows with many integration steps.

- **Seamless SMC integration.** FP++ naturally integrates with Sequential Monte Carlo (SMC) annealing (Liu & Chen, 1998), supporting stable Boltzmann reweighting and mitigating particle degeneracy.

- **Scalable validation.** Experiments on a 1000-dimensional Gaussian mixture and an all-atom Chignolin mutant show FP++ improves equilibrium sampling over Hutchinson and original FP baselines.

## 2. Related Work

### 2.1. Balancing Expressivity and Tractability in Flows

Generative models for Boltzmann sampling differ primarily in how they handle the Jacobian determinant.

**Constrained architectures.** Early flow models such as NICE (Dinh et al., 2015), RealNVP (Dinh et al., 2017), and Glow (Kingma & Dhariwal, 2018) use structured transformations (e.g., affine coupling) that yield triangular Jacobians and allow $\mathcal{O}(D)$ determinant evaluation. While computationally efficient, these architectural constraints limit expressivity and are often insufficient for complex molecular or high-dimensional target distributions (Noé et al., 2019).

**Free-form continuous flows.** More flexible approaches—including Neural ODEs (Chen et al., 2018), diffusion-based Probability Flows (Ho et al., 2020; Song et al., 2021), flow-matching methods (Lipman et al., 2023), and i-ResNets (Behrmann et al., 2019)—remove structural constraints to model richer dynamics. i-ResNets enforce invertibility via residual blocks and estimate determinant product with Hutchinson's trace estimator, while diffusion and flow-matching flows learn vector fields without explicit Jacobian computations during training. However, exact likelihood and unbiased inference still require evaluating Jacobian determinants, leaving a fundamental trade-off between expressivity and tractability.

### 2.2. Sequential Monte Carlo

High-dimensional importance sampling often suffers from weight collapse. Sequential Monte Carlo (SMC) (Liu & Chen, 1998) mitigates this via tempering and resampling. Flow-SMC approaches (Arbel et al., 2021; Speich et al., 2021) combine learned flows with SMC to stabilize weight evolution. FP++ integrates naturally with SMC annealing, providing unbiased and variance-reduced Boltzmann reweighting in high dimensions.

## 3. Background

### 3.1. Boltzmann Reweighting

Given a generative model $q(\mathbf{z})$ induced by a mapping $\mathbf{f} : \mathbf{z} \mapsto \mathbf{x}$ $(\mathbf{z} \sim q(\mathbf{z}))$, the unbiased Boltzmann expectation of an observable $O(\mathbf{x})$ is

$$\mathbb{E}_p[O] = \mathbb{E}_{\mathbf{z} \sim q(\mathbf{z})} \Big[ O(\mathbf{f}(\mathbf{z})) \, e^{-W(\mathbf{z})} \Big], \qquad (1)$$

where the generalized work $W$ is

$$W(\mathbf{z}) = u_X(\mathbf{f}(\mathbf{z})) - u_Z(\mathbf{z}) - \Delta S(\mathbf{z}), \qquad (2)$$

with $u_X(\mathbf{x})$ and $u_Z(\mathbf{z})$ denoting the dimensionless potentials of the target Boltzmann and prior distributions, respec-

tively. The entropic contribution

$$\Delta S(\mathbf{z}) = \log \left| \det \left( \frac{\partial \mathbf{f}}{\partial \mathbf{z}} \right) \right| \tag{3}$$

is determined by the Jacobian of the generative map and represents the primary computational bottleneck. In practice, the importance weights $e^{-W(\mathbf{z})}$ are normalized.

## 3.2. Probability Flow ODEs

Diffusion-based generative models define an SDE

$$d\mathbf{x}_t = f(\mathbf{x}_t, t)\, dt + g(t)\, d\mathbf{w}_t. \tag{4}$$

The corresponding probability flow ODE (Song et al., 2021) is

$$\frac{d\mathbf{x}_t}{dt} = f(\mathbf{x}_t, t) - \frac{1}{2} g(t)^2 \nabla_{\mathbf{x}} \log p_t(\mathbf{x}_t), \tag{5}$$

which defines a special type of continuous normalizing flow (CNF). During training, score-matching allows the model to avoid direct computation of the Jacobian determinant, which would be prohibitively expensive in high dimensions.

At inference time, evaluating importance weights or likelihoods requires the log-determinant of the flow Jacobian (Köhler et al., 2020):

$$\log \left| \det \left( \frac{\partial \mathbf{f}}{\partial \mathbf{z}} \right) \right| = \int_0^T \nabla_{\mathbf{x}} \cdot v(\mathbf{x}_t, t)\, dt, \tag{6}$$

where $v(\mathbf{x}_t, t) = f(\mathbf{x}_t, t) - \frac{1}{2} g(t)^2 \nabla_{\mathbf{x}} \log p_t(\mathbf{x}_t)$. Here, $\nabla_{\mathbf{x}} \cdot v$ denotes the divergence of the velocity field, which can be obtained as the trace of its Jacobian. This term is computationally expensive in high dimensions, which motivates estimating the divergence using the Hutchinson estimator (see Appendix D).

## 3.3. Jacobian Determinant Estimation

Continuous flows require Jacobian determinant for likelihood evaluation or importance sampling. Hutchinson's estimator (Hutchinson, 1989) approximates traces via randomized projections,

$$\hat{T}_{\text{Hutch}} = \mathbf{u}^\top \left( \frac{\partial v(\mathbf{x})}{\partial \mathbf{x}} \right) \mathbf{u}, \quad \mathbf{u} \sim \mathcal{D}, \ \mathbb{E}[\mathbf{u}\mathbf{u}^\top] = \mathbf{I}, \tag{7}$$

but it is not an unbiased estimator of the Jacobian determinant and exhibits significant bias in high-dimensional flows (Behrmann et al., 2019; Noé et al., 2019).

To alleviate the computational cost of evaluating the flow Jacobian at inference, Flow Perturbation (FP) (Peng & Gao, 2025) provides an unbiased estimator for the Jacobian determinant:

$$\hat{\mathcal{J}}_{\text{FP}} = \left\| \frac{\partial \mathbf{f}^{-1}}{\partial \mathbf{x}} \epsilon \right\|^{-D}, \quad \epsilon \sim \mathbb{S}^{D-1}, \tag{8}$$

where $\epsilon$ is sampled uniformly from the unit sphere in $D$ dimensions.

## 4. Method

### 4.1. Flow Perturbation++ Estimator

Numerical integration of the ODE from $t = 0$ to $t = T_{max}$ decomposes the flow $\mathbf{f}$ into $T$ discrete steps:

$$\mathbf{f} = \mathbf{f}_1 \circ \mathbf{f}_2 \circ \cdots \circ \mathbf{f}_T, \quad \det \left( \frac{\partial \mathbf{f}}{\partial \mathbf{z}} \right) = \prod_{k=1}^{T} \det(J_k), \tag{9}$$

where $J_k = \partial \mathbf{f}_k / \partial \mathbf{z}_k$.

**FP++ Estimator.** As noted in FP (Peng & Gao, 2025), estimating the Jacobian can suffer from high variance due to correlations in single-step estimators. FP++ reduces this variance by applying independent perturbations at each integration step:

$$\hat{\mathcal{J}}_{\text{FP++}} = \prod_{k=1}^{K} \| J_k^{-1} \epsilon_k \|^{-D}, \quad \epsilon_k \overset{\text{i.i.d.}}{\sim} \mathbb{S}^{D-1}, \tag{10}$$

where $J_k^{-1} = \partial \mathbf{f}_k^{-1} / \partial \mathbf{x}_k$ maps perturbations from the output of the $k$-th step back to its input, capturing the local volume change at that step. This yields the entropic contribution:

$$\Delta \hat{S}_{\text{FP++}} = \log \hat{\mathcal{J}}_{\text{FP++}}. \tag{11}$$

Importantly, FP++ is an unbiased estimator of the full flow Jacobian:

$$\mathbb{E}[\hat{\mathcal{J}}_{\text{FP++}}] = \left| \det \frac{\partial \mathbf{f}}{\partial \mathbf{z}} \right|, \tag{12}$$

with a formal proof provided in Appendix A.

The inverse–Jacobian–vector product $J_k^{-1} \epsilon_k$ appearing in Eq. (10) can be approximated via finite differences:

$$J_k^{-1} \epsilon_k \approx \frac{\mathbf{f}_k^{-1}(\mathbf{x}_k + \delta \epsilon_k) - \mathbf{f}_k^{-1}(\mathbf{x}_k - \delta \epsilon_k)}{2\delta}, \tag{13}$$

or exactly via automatic differentiation with VJP or JVP products (see Appendix E for details).

**Variance Reduction.** FP++ achieves lower variance than full-flow FP by using independent perturbations at each step, eliminating correlations that inflate variance in single-step FP. Detailed derivation is in Appendix B.

### 4.2. Integration with Sequential Monte Carlo

In high-dimensional systems, importance weights often concentrate on only a few samples. We address this using SMC with geometric annealing:

$$\pi_n(\mathbf{z}) \propto p(\mathbf{z})^{\beta_n} q(\mathbf{z})^{1-\beta_n}, \quad 0 = \beta_0 < \cdots < \beta_N = 1, \tag{14}$$

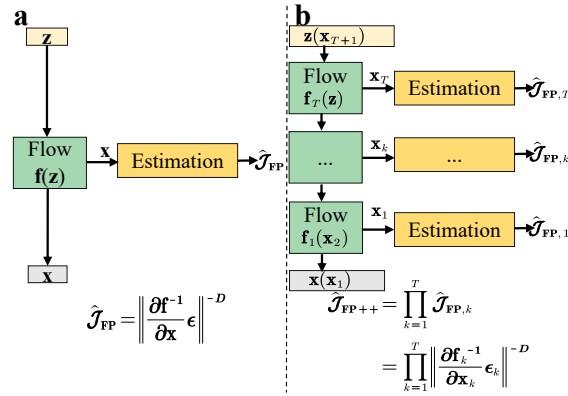

*Figure 1.* **(a)** Single-step Flow Perturbation (FP) computes the full flow Jacobian in one shot, often resulting in high variance. **(b)** Flow Perturbation++ (FP++) breaks the flow into $T$ steps, applying unbiased estimation sequentially from latent $\mathbf{z} = \mathbf{x}_{T+1}$ to the final sample $\mathbf{x} = \mathbf{x}_1$. The product of per-step Jacobians gives a low-variance, numerically stable approximation of the full determinant.

---

**Algorithm 1** SMC with Flow Perturbation++ (FP++)

---

**Input:** Prior $\pi_0$, target $\pi_T$, annealing schedule $\{\beta_n\}_{n=0}^N$

**Initialize:** $\{\mathbf{z}_0^{(i)}\}_{i=1}^M \sim \pi_0$, $\omega_0^{(i)} = 1/M$

**for** $n = 1$ **to** $N$ **do**

    **(a) MCMC Propagation:** $\mathbf{z}_n^{(i)} \sim K_n(\mathbf{z}_{n-1}^{(i)}, \cdot)$ *(cost included in Table 4)*

    **(b) Work Increment:** Compute $W^{(i)}$ using $\Delta S_n^{(i)}$ and $u(\mathbf{x})$

    **(c) Weight Update:** $\omega_n^{(i)} \propto \omega_{n-1}^{(i)} \exp\big(-(\beta_n - \beta_{n-1})W^{(i)}\big)$

    **if** ESS < threshold **then**

        **(d) Resample:** Resample particles

    **end if**

**end for**

**Output:** Weighted particles $\{\mathbf{z}_N^{(i)}, \omega_N^{(i)}\}$

---

and update incremental weights via

$$\omega_n(\mathbf{z}) = \omega_{n-1}(\mathbf{z}) \exp\big(-(\beta_n - \beta_{n-1})W(\mathbf{z})\big), \quad (15)$$

where $W(\mathbf{z})$ is the generalized work estimated by FP++. Resampling is triggered when

$$\text{ESS} = \frac{1}{\sum_j (\omega_n^{(j)})^2} \quad (16)$$

falls below a threshold. The full FP++–SMC procedure is summarized in Algorithm 1.

### 4.3. Computational Complexity Comparison

We compare the computational cost of baseline Jacobian-determinant estimators with the proposed FP++ (multi-step) approach. Baselines include single-step Flow Perturbation

(FP) (Peng & Gao, 2025), the Hutchinson trace estimator (Hutchinson, 1989), and the exact brute-force Jacobian (BFJacob). Table 1 summarizes their key characteristics of these methods. All methods require one forward ODE integration to generate the trajectory $\mathbf{x} = f(\mathbf{z})$; additional cost stems from estimating the entropic term.

- **BFJacob** $(1 + D)$**:** Computes the full $D \times D$ Jacobian via $D$ vector–Jacobian products (VJPs), each requiring a separate backward pass along the flow. The total cost corresponds to 1 ODE pass for sample generation plus $D$ passes for the VJPs, i.e., $1 + D$.

- **Hutchinson** $(1 + N)$**:** Estimates the trace using $N$ random probes, each involving one VJP along the flow. The total cost is 1 (generation) $+N$ (VJPs), with small $N$ potentially leading to high variance or bias.

- **FP / FP++** $(3)$**:** Estimates $\det(J)$ from $J^{-1}\epsilon$ using central finite differences (Equation (13)), requiring two inverse flow evaluations along the trajectory. The total cost is fixed at 1 (generation) $+2$ (estimation) $= 3$, independent of $D$.

*Table 1.* Computational cost comparison of baseline Jacobian-determinant methods and FP++ (multi-step). Cost is measured by the number of ODE passes (trajectory + entropic computation).

| Method | Jacobian Calc. | Biased | Cost (# passes) |
|---|---|---|---|
| FP / FP++ | No | No | 3 |
| BFJacob | Yes | No | $1 + D$ |
| Hutch | No | Yes | $1 + N$ |

## 5. Experiments and Results

We evaluate the FP++-SMC framework on two representative high-dimensional systems: a synthetic Gaussian Mixture Model (GMM), and the all-atom Chignolin peptide.

### 5.1. Experimental Setup

#### 5.1.1. TEST SYSTEMS

**Gaussian Mixture Model (GMM).** We consider a high-dimensional two-component anisotropic Gaussian mixture as a benchmark system. Training samples are drawn uniformly from the mixture, while the target Boltzmann distribution assigns asymmetric component weights $(0.25, 0.75)$, making it challenging for models to fully match the target distribution. This bimodal system is structurally simple, yet poses a nontrivial sampling challenge due to metastability between modes, and admits an analytically tractable reference solution, making it an ideal platform for evaluating the performance of FP++. Full system specifications are provided in Appendix C.

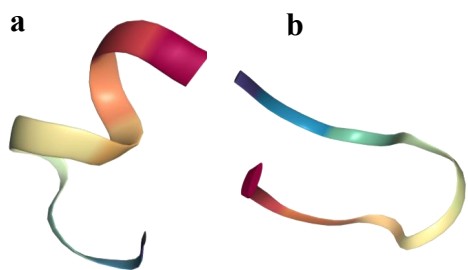

*Figure 2.* Representative CNL025 configurations at 300 K: (a) partially $\alpha$-helical; (b) folded $\beta$-hairpin.

**Chignolin Mutant CNL025.** The 10-residue Chignolin mutant CNL025 (Rodriguez et al., 2011; McKiernan et al., 2017; Maruyama & Mitsutake, 2018) provides a compact and challenging biomolecular test case. We adopt the CHARMM22 force field (MacKerell Jr. et al., 2000) with the OBC2 implicit solvent model (Onufriev et al., 2004). Training configurations at 300 K are extracted from replica-exchange molecular dynamics data released by the Noé group.[1] Representative conformations are shown in Fig. 2.

### 5.1.2. FLOW MODELS

For all experiments, we use the pretrained continuous-time flow models released in (Peng & Gao, 2025) without further training. The models were trained with the $v$-prediction objective (Salimans & Ho, 2022), which improves numerical stability in high dimensions, with the neural network representing the time-dependent velocity field $v_\theta(\mathbf{x}, t)$ that defines the probability flow. Full architecture and training details are reported in (Peng & Gao, 2025).

For the GMM benchmark, a lightweight multilayer perceptron is sufficient to model the low-complexity vector field associated with the mixture structure. For Chignolin, we employ a Transformer-based architecture (Vaswani et al., 2017; Li et al., 2022) to parameterize $v_\theta(\mathbf{x}, t)$, enabling the model to capture long-range geometric and many-body correlations in high-dimensional biomolecular configurations.

### 5.1.3. COMPARED JACOBIAN ESTIMATORS

We evaluate our multi-step **FP++** estimator against three baselines under identical SMC pipelines:

- **FP (single-step)** (Peng & Gao, 2025), the original perturbation-based estimator.

- **Hutchinson trace estimators** (Hutchinson, 1989), using Gaussian or Rademacher probes, denoted as 'G$n$' and 'R$n$' respectively, with $n \in \{1, 2, 10\}$.

[1]http://ftp.mi.fu-berlin.de/pub/cmbdata/bgmol/datasets/chignolin/ChignolinOBC2PT.tgz

- **Brute-force Jacobian (BFJacob)**, an exact but computationally expensive method via automatic differentiation.

### 5.1.4. EVALUATION METRICS

We report several standard metrics to quantify estimator accuracy and sampling performance.

- **Energy Distribution.** We compare empirical SMC energy histograms with the reference Boltzmann distribution.

- **Computational Cost.** We measure the approximate wall-clock time required for one full SMC run.

- **Reaction-Coordinate Occupancy.** We compute occupancy along a selected reaction coordinate to evaluate recovery of the free-energy landscape.

- **Remaining Ancestor Count.** We track the number of surviving ancestors after resampling (Del Moral et al., 2006), a diagnostic indicator of particle degeneracy.

## 5.2. Results

### 5.2.1. LOW-DIMENSIONAL GMM (10D)

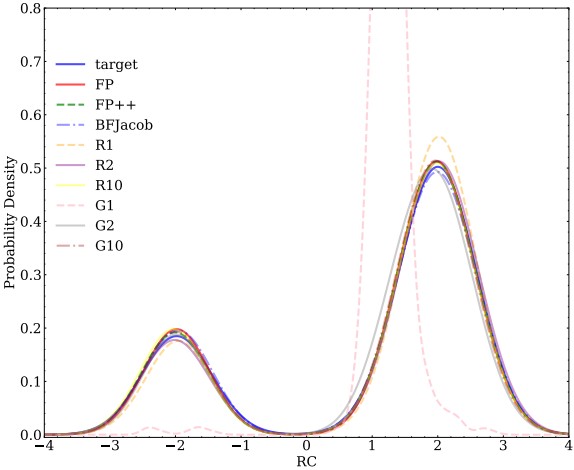

*Figure 3.* Final SMC sample distribution along the reaction coordinate for the 10D GMM. The solid blue curve shows the target distribution obtained via direct sampling.

We assess estimator performance on a 10-dimensional Gaussian Mixture Model using SMC with 1,000 particles across 10 independent runs. Accuracy is measured by the fraction of particles assigned to the first Gaussian component. As shown in Table 2, both the FP baseline and FP++ accurately recover the target modal probability (25%), matching the ground truth. Hutchinson estimators exhibit noticeable bias when using only a few random probe vectors ($n = 1, 2$),

with accuracy improving as the number of probe noises increases. The brute-force Jacobian ("BF Jacobian") provides high accuracy but at substantially greater computational cost. The reaction-coordinate distributions in Fig. 3 further demonstrate that FP++ yields stable and accurate reconstruction of the target distribution.

*Table 2.* Performance of baseline and proposed estimators on the 10D GMM. The modal weight reports the fraction of SMC particles located in the first Gaussian component.

| Estimator | Mean | Std | # Preserved | Time |
|---|---|---|---|---|
| Hutchinson G1 | 0.0189 | 0.018 | 131 | 2.5m |
| Hutchinson G2 | 0.2278 | 0.013 | 828 | 3.0m |
| Hutchinson G10 | 0.2454 | 0.011 | 943 | 3.0m |
| Hutchinson R1 | 0.2289 | 0.017 | 972 | 2.5m |
| Hutchinson R2 | 0.2391 | 0.010 | 971 | 3.0m |
| Hutchinson R10 | 0.2503 | 0.016 | 971 | 3.0m |
| BF Jacobian | 0.2497 | 0.019 | 978 | 5.5m |
| FP (single-step) | 0.2499 | 0.015 | 862 | 2.5m |
| **FP++** (multi-step) | 0.2503 | 0.011 | 957 | 2.5m |

### 5.2.2. 1000 DIMENSIONAL GMM

We further benchmark all estimators on a substantially more challenging 1000-dimensional GMM. For FP, FP++, and Hutchinson variants, SMC is performed with 1,000 particles and ten independent runs, using 2,000 annealing steps with 10 MCMC transitions per step. Due to its prohibitive cost, the Brute-Force Jacobian baseline (BFJacob) is evaluated using 100 particles in a single run.

*Table 3.* Performance of Jacobian estimators on the 1000-dimensional GMM. Modal weights report the fraction of particles assigned to the first Gaussian component.

| Estimator | Mean | Std | # Preserved | Time |
|---|---|---|---|---|
| Hutchinson G1 | 0.534 | 0.073 | 154 | 0.7 h |
| Hutchinson G2 | 0.377 | 0.023 | 222 | 1.1 h |
| Hutchinson G10 | 0.277 | 0.022 | 360 | 5.1 h |
| Hutchinson R1 | 0 | 0 | 328 | 0.7 h |
| Hutchinson R2 | 0 | 0 | 361 | 1.1 h |
| Hutchinson R10 | 0.044 | 0.011 | 409 | 5.1 h |
| BF Jacobian | 0.400 | - | 45 | 31 h |
| FP (single-step) | 0.209 | 0.083 | 59 | 1.1 h |
| **FP++** (multi-step) | 0.256 | 0.027 | 312 | 1.1 h |

Table 3 reports the estimated probability of the first mode (ground truth 25%), lineage preservation, and computational cost. Among all methods, **FP++ best recovers the target modal probability**, yielding $0.256 \pm 0.027$ across ten runs while preserving a large number of lineages (312 on aver-

age). Notably, FP++ achieves this accuracy at essentially the same computational cost as the single-step FP estimator.

Using 10 Gaussian Hutchinson probes (G10) attains the next-best performance ($0.277 \pm 0.022$), but requires roughly five times the computational cost. The single-step FP estimator produces reasonable estimates ($0.209 \pm 0.083$) with substantially higher variance. All other Hutchinson variants suffer from significant bias or mode collapse, accompanied by severe lineage degeneracy. BFJacob underperforms at this scale due to the limited particle budget; extrapolating to 1,000 particles would increase its computational cost by roughly 300× compared with FP++, making it entirely impractical.

Figure 4 shows final sample and energy distributions (a,d) and acceptance rates and average energies across the annealing schedule (b,c). FP++ consistently matches or exceeds the accuracy of both G10 and BFJacob while using only a fraction of their computational budget. Partial coordinate updates induce stepwise acceptance-rate fluctuations, with FP exhibiting the highest variance and lowest acceptance rates. FP++ reduces this variance by per-step Jacobian decomposition, resulting in consistently higher acceptance rates.

### 5.2.3. CHIGNOLIN MUTANT

*Table 4.* Performance of different Jacobian estimators on the Chignolin mutant. The table reports the fraction of particles located in the right basin of Figure 5, corresponding to the $\beta$-hairpin configuration, along with the number of preserved lineages and the total computational cost of each method. The target value, derived from the training data, is 0.385; direct sampling from the base flow yields 0.311.

| Method | Proportion | # Preserved | Time |
|---|---|---|---|
| Hutchinson R2 | 0.225 | 1537 | 30.4 h |
| Hutchinson R10 | 0.273 | 1601 | 100 h |
| Hutchinson G2 | 0.195 | 1330 | 30.4 h |
| Hutchinson G10 | 0.222 | 1445 | 100 h |
| FP (single-step) | 0.475 | 1428 | 11.1 h |
| **FP++** (multi-step) | 0.364 | 1463 | 11.1 h |

For the Chignolin mutant, we conducted SMC simulations using the FP, FP++, and Hutchinson estimators. Each simulation employed 2,000 particles, 600 intermediate distributions, and 2 MCMC steps per level. Resampling was performed using the KL Reshuffling scheme (Kviman et al., 2022). The Brute-Force Jacobian baseline (BFJacob) was omitted due to its prohibitive computational cost: even a single run with a reduced number of particles would already require orders of magnitude more resources than FP++ or Hutchinson estimators, making it infeasible for this system.

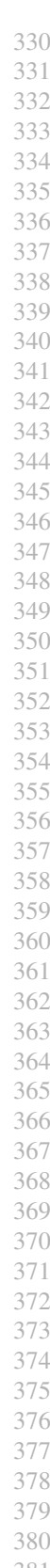

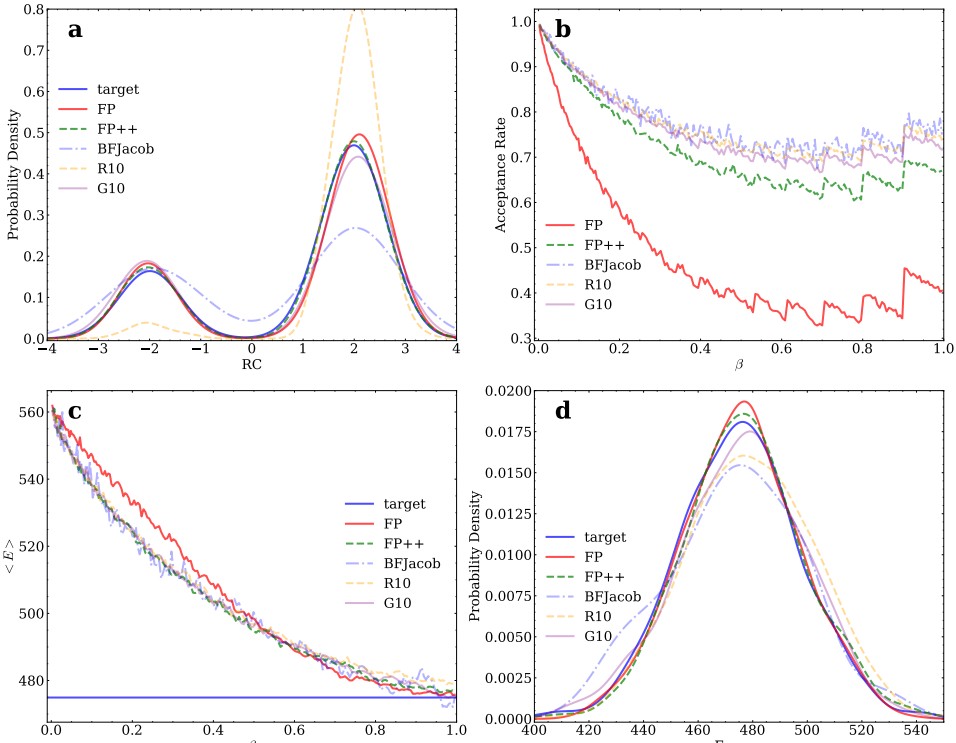

*Figure 4.* SMC results for the 1000-dimensional GMM. (a) Probability distribution of final SMC samples along the reaction coordinate. (b) Monte Carlo acceptance rate as a function of $\beta$. (c) Average sample energy as a function of $\beta$. (d) Final energy distribution. The target energy and distribution (solid line) are obtained by direct sampling from the target GMM. All results are derived from a single SMC run.

Table 4 reports the fraction of particles occupying the right (folded) basin, with a target occupancy of 0.385. Among all tested estimators, **FP++ achieves the closest agreement with the target** (0.364), followed by the single-step FP estimator (0.475), both evaluated under the same computational budget. Hutchinson-based estimators systematically underestimate basin occupancy, even with 10 Gaussian (G10) or Rademacher (R10) probe vectors, highlighting the limitations of single-vector or low-probe approximations. Variants with fewer probes (G2/R2) yield intermediate results but remain less accurate than FP++, emphasizing the importance of multi-step variance reduction.

Although finite-difference approximations incur a comparable number of model evaluations to low-probe Hutchinson estimators (e.g., G2/R2), they can be substantially faster in practice for Transformer-based flows. As shown in Table 4, finite-difference–based FP and FP++ achieve significantly lower wall-clock times than Hutchinson-type estimators under comparable or even larger computational budgets. This behavior arises from the use of self-attention and normalization layers: Jacobian–vector products require backpropagation through these components, whose computational cost is dominated by dense attention operations and softmax derivatives. Such backward passes are well known to be con-

siderably more expensive than forward evaluations (Vaswani et al., 2017; Liang et al., 2025). In contrast, finite-difference estimators only involve multiple forward passes, which are heavily optimized on modern GPUs via fused kernels and specialized attention implementations. For consistency, Appendix E reports runtimes with all methods implemented using VJP or JVP, including FP and FP++.

Figure 5 illustrates the projected particle distributions along the reaction-coordinate plane. Both FP and FP++ faithfully recover the bimodal target distribution, with negligible leakage into the low-probability corridor separating the $\alpha$-helical and $\beta$-hairpin basins. In contrast, Hutchinson-based and base-flow samples display substantial spurious occupancy between basins.

Energy histograms (Figure 5i) further confirm that FP and FP++ generate high-fidelity ensembles compared to Hutchinson estimators or direct base-flow samples. Acceptance rates along the annealing schedule (Figure 5g) remain smooth, consistent with largely fixed-coordinate updates, and FP++ exhibits a slight improvement over FP, reflecting reduced variance through the multi-step decomposition.

For reference, conventional MD trajectories often become trapped in single metastable states, underscoring the chal-

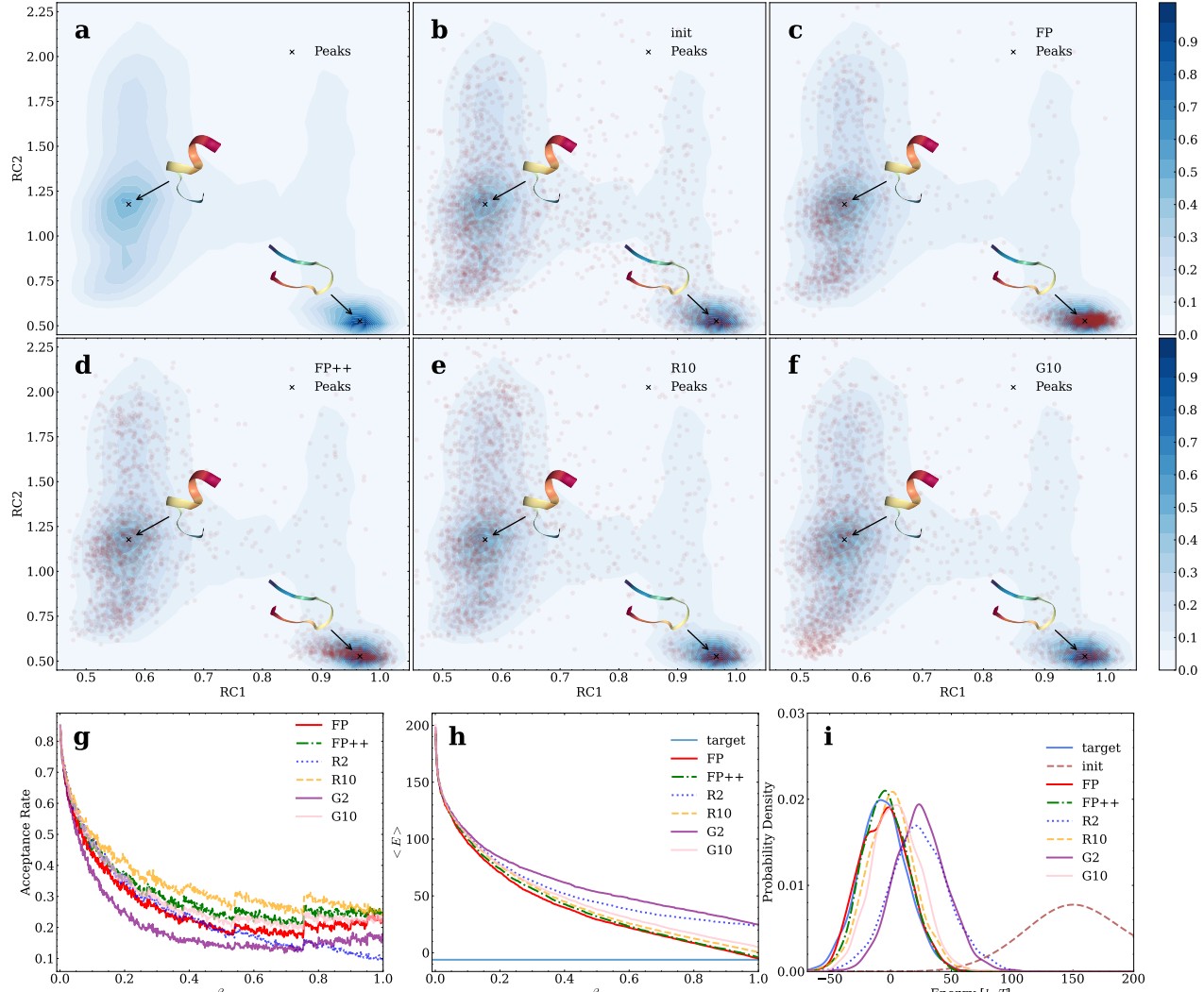

*Figure 5.* SMC results for the Chignolin mutant. (a–f) Sample distributions projected onto the reaction-coordinate (RC) plane: (a) Target distribution derived from the training data; (b) Samples generated directly by the base flow model; (c) Samples produced by SMC using the single-step FP estimator; (d) Samples produced by SMC using the FP++ estimator; (e) Samples produced by SMC with the Hutchinson estimator (10 Rademacher vectors); (f) Samples produced by SMC with the Hutchinson estimator (10 Gaussian vectors). (g) Monte Carlo acceptance rates as functions of $\beta$; (h) Average energy $\langle E \rangle$ of intermediate SMC samples; (i) Energy distributions at the end of SMC compared to the target distribution and the base flow ("init"). All SMC results are derived from a single run.

lenges of effectively sampling the Chignolin mutant using standard simulation approaches (Appendix F).

## 6. Conclusion

We have introduced **Flow Perturbation++ (FP++)**, a theoretically grounded, unbiased, and variance-reduced estimator of the flow Jacobian determinant that leverages the semigroup property of flow maps to decompose the global Jacobian into per-step determinants. FP++ serves as a practical and scalable alternative to Hutchinson-style trace estima-

tors (Hutchinson, 1989). When integrated with sequential Monte Carlo (SMC), FP++ enables accurate Boltzmann reweighting and reliable thermodynamic statistics across both synthetic high-dimensional Gaussian mixtures and realistic biomolecular systems such as the Chignolin mutant. Our experiments demonstrate that FP++ not only improves estimator accuracy and stability compared to both the original FP and Hutchinson estimators, but also substantially reduces computational cost relative to conventional brute-force Jacobian evaluation.

## Software and Data

To support reproducibility, we provide an anonymous implementation of all methods described in this paper, along with scripts for data generation and evaluation. The code is publicly available at the following anonymous repository:

https://anonymous.4open.science/r/Flow_
Perturbation_pp-9EC5

The repository will be updated with a fully documented and permanent version upon acceptance.

## Impact Statement

FP++ provides a general-purpose, unbiased alternative to Hutchinson trace estimators for computing Jacobian determinants in continuous generative flows. By reducing variance and supporting multi-noise extensions, it enables rigorous thermodynamic inference and Boltzmann reweighting from pretrained generative models. This work advances the use of flow-based generative models in scientific applications, including molecular simulations and high-dimensional inference, without raising immediate ethical concerns.

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

# A. Proof of Flow Perturbation++

In this appendix we provide a complete derivation of the Flow Perturbation++ (FP++) estimator. Consider a flow map $\mathbf{f} : \mathbb{R}^D \to \mathbb{R}^D$ obtained by composing $T$ numerical ODE integration steps:

$$\mathbf{f} = \mathbf{f}_1 \circ \mathbf{f}_2 \circ \cdots \circ \mathbf{f}_T.$$

For an initial state $\mathbf{z} \sim q(\mathbf{z})$ we define a trajectory

$$\mathbf{x}_1 = \mathbf{x}, \quad \mathbf{x}_i = \mathbf{f}_i(\mathbf{x}_{i+1}), \quad \mathbf{x}_{T+1} = \mathbf{z}.$$

Let $\hat{\epsilon}_1, \ldots, \hat{\epsilon}_T$ be i.i.d. random vectors sampled uniformly from the unit sphere $S^{D-1}$. Define the FP++ action

$$\Delta S(\mathbf{z}, \hat{\epsilon}_1, \ldots, \hat{\epsilon}_T) = -D \sum_{i=1}^{T} \log \left\| \frac{\partial \mathbf{f}_i^{-1}}{\partial \mathbf{x}_i} \hat{\epsilon}_i \right\|.$$

We aim to show that for any target density $p(x)$, prior density $q(z)$, and observable $O(x)$,

$$\mathbb{E}_{\mathbf{x} \sim p(\mathbf{x})}[O(\mathbf{x})] = \mathbb{E}_{\mathbf{z} \sim q(\mathbf{z}), \hat{\epsilon}_{1:T}} \left[ O(\mathbf{f}(\mathbf{z})) \frac{p(\mathbf{f}(\mathbf{z}))}{q(\mathbf{z})} \exp\big(\Delta S(\mathbf{z}, \hat{\epsilon}_{1:T})\big) \right]. \tag{17}$$

This identity holds for every test function $O$, hence it suffices to show that

$$\mathbb{E}_{\hat{\epsilon}_{1:T}} \left[ \exp\big(\Delta S(\mathbf{z}, \hat{\epsilon}_{1:T})\big) \right] = \left| \det \frac{\partial \mathbf{f}}{\partial \mathbf{z}} \right|. \tag{18}$$

## A.1. Reduction to a Single-Step Identity

Expanding the FP++ weight gives

$$\exp(\Delta S) = \prod_{i=1}^{T} \left\| \frac{\partial \mathbf{f}_i^{-1}}{\partial \mathbf{x}_i} \hat{\epsilon}_i \right\|^{-D}.$$

Because the $\hat{\epsilon}_i$ are independent,

$$\mathbb{E}_{\hat{\epsilon}_{1:T}}[\exp(\Delta S)] = \prod_{i=1}^{T} \mathbb{E}_{\hat{\epsilon}_i} \left[ \left\| \frac{\partial \mathbf{f}_i^{-1}}{\partial \mathbf{x}_i} \hat{\epsilon}_i \right\|^{-D} \right].$$

Thus, the main statement reduces to proving the following single-step identity:

$$\mathbb{E}_{\hat{\epsilon} \sim S^{D-1}} \left[ \|A\hat{\epsilon}\|^{-D} \right] = |\det(A)|^{-1}, \qquad A \in \mathbb{R}^{D \times D} \text{ invertible.} \tag{19}$$

## A.2. Proof of the Single-Step Identity

Let $A = U\Sigma V^\top$ be the SVD of $A$, where $U$ and $V$ are orthogonal matrices and $\Sigma = \mathrm{diag}(\sigma_1, \ldots, \sigma_D)$ contains the singular values of $A$. Orthogonal matrices preserve the uniform measure on the unit sphere, therefore $V^\top \hat{\epsilon}$ is uniformly distributed on $S^{D-1}$ as well. Using the rotational invariance of the Euclidean norm:

$$\|A\hat{\epsilon}\| = \|U\Sigma V^\top \hat{\epsilon}\| = \|\Sigma v\|, \qquad v = V^\top \hat{\epsilon}.$$

Thus,

$$\mathbb{E}_{\hat{\epsilon}}[\|A\hat{\epsilon}\|^{-D}] = \mathbb{E}_{v \sim S^{D-1}} \left[ \left( \sum_{j=1}^{D} \sigma_j^2 v_j^2 \right)^{-D/2} \right].$$

A classical result from integral geometry (obtainable via a change to ellipsoidal coordinates or by comparing Gaussian integrals in two ways) states that

$$\mathbb{E}_{v \sim S^{D-1}} \left[ \left( \sum_{j=1}^{D} \sigma_j^2 v_j^2 \right)^{-D/2} \right] = \frac{1}{\prod_{j=1}^{D} \sigma_j}. \tag{20}$$

Since $\prod_{j=1}^{D} \sigma_j = |\det(A)|$, equation (19) follows immediately.

### A.3. Completion of the FP++ Proof

Applying (19) to each integration step with $A_i = \frac{\partial \mathbf{f}_i^{-1}}{\partial \mathbf{x}_i}$ yields

$$\mathbb{E}_{\hat{\epsilon}_{1:T}}[\exp(\Delta S)] = \prod_{i=1}^{T} \left| \det\left(\frac{\partial \mathbf{f}_i^{-1}}{\partial \mathbf{x}_i}\right) \right|^{-1} = \prod_{i=1}^{T} \left| \det\left(\frac{\partial \mathbf{f}_i}{\partial \mathbf{x}_{i+1}}\right) \right|.$$

By the chain rule for Jacobians,

$$\prod_{i=1}^{T} \left| \det\left(\frac{\partial \mathbf{f}_i}{\partial \mathbf{x}_{i+1}}\right) \right| = \left| \det\left(\frac{\partial \mathbf{f}}{\partial \mathbf{z}}\right) \right|.$$

Substituting into (17) yields the FP++ importance sampling identity, completing the proof. □

## B. Flow Perturbation++ Variance Reduction Analysis

In this section, we provide a rigorous mathematical explanation for why the variance of the Flow Perturbation++ (FP++) estimator is smaller than that of the full-flow Flow Perturbation (FP) estimator.

Recall that in FP++, we use independent random perturbations for each flow step, whereas in FP, the same perturbation is propagated through all flow steps. For simplicity, denote the contribution of the $i$-th step as

$$X_i = -D \log \left\| \frac{\partial \mathbf{f}_i^{-1}}{\partial \mathbf{x}_i} \hat{\epsilon}_i \right\|, \tag{21}$$

where $\hat{\epsilon}_i$ is the unit random perturbation at step $i$ in FP++, and

$$Y_i = -D \log \left\| \frac{\partial \mathbf{f}_i^{-1}}{\partial \mathbf{x}_i} \hat{\epsilon} \right\| \tag{22}$$

is the corresponding contribution in FP using a single shared $\hat{\epsilon}$ for all steps. The total contributions are

$$\Delta S_{\text{FP++}} = \sum_i X_i, \quad \Delta S_{\text{FP}} = \sum_i Y_i. \tag{23}$$

### B.1. Single-Step Variance

Since each flow step acts on a random unit vector with identical statistical properties, the single-step variances are exactly equal:

$$\text{Var}[X_i] = \text{Var}[Y_i]. \tag{24}$$

### B.2. Covariance Structure

The total variance for FP is

$$\text{Var}\left[\sum_i Y_i\right] = \sum_i \text{Var}[Y_i] + 2 \sum_{i<j} \text{Cov}[Y_i, Y_j]. \tag{25}$$

Consider the contributions at steps $i$ and $j$ in full-flow FP:

$$Y_i = -D \log \|J_i \hat{\epsilon}\|, \quad Y_j = -D \log \|J_j \hat{\epsilon}\|, \tag{26}$$

where $\hat{\epsilon}$ is a random unit vector shared across steps, and $J_i = \partial \mathbf{f}_i^{-1}/\partial x_i$.

The covariance is

$$\text{Cov}[Y_i, Y_j] = \mathbb{E}[Y_i Y_j] - \mathbb{E}[Y_i]\mathbb{E}[Y_j]. \tag{27}$$

Define

$$g_i(\hat{\epsilon}) = \log \|J_i \hat{\epsilon}\|, \quad g_j(\hat{\epsilon}) = \log \|J_j \hat{\epsilon}\|. \tag{28}$$

Then

$$\mathrm{Cov}[Y_i, Y_j] = D^2 \, \mathrm{Cov}[g_i(\hat{\epsilon}), g_j(\hat{\epsilon})]. \tag{29}$$

For a unit vector $\hat{\epsilon}$, we can write

$$g_i(\hat{\epsilon}) = \frac{1}{2} \log(\hat{\epsilon}^\top A_i \hat{\epsilon}), \quad g_j(\hat{\epsilon}) = \frac{1}{2} \log(\hat{\epsilon}^\top A_j \hat{\epsilon}), \tag{30}$$

where $A_i = J_i^\top J_i$ and $A_j = J_j^\top J_j$ are symmetric positive semi-definite matrices.

**Justification in the FP context.** In practice, for continuous normalizing flows, consecutive Jacobian matrices $J_i$ and $J_j$ correspond to small integration steps. Hence, the dominant eigenvectors of $A_i$ and $A_j$ are highly aligned, and the functions $\hat{\epsilon} \mapsto \hat{\epsilon}^\top A_i \hat{\epsilon}$ and $\hat{\epsilon} \mapsto \hat{\epsilon}^\top A_j \hat{\epsilon}$ are approximately co-monotonic along most directions sampled by $\hat{\epsilon}$. Under this condition, the covariance is typically non-negative:

$$\mathrm{Cov}[g_i(\hat{\epsilon}), g_j(\hat{\epsilon})] \geq 0. \tag{31}$$

Thus,

$$\mathrm{Cov}[Y_i, Y_j] = D^2 \, \mathrm{Cov}[g_i(\hat{\epsilon}), g_j(\hat{\epsilon})] \geq 0. \tag{32}$$

This shows rigorously that the covariance of step contributions in full-flow FP is non-negative. Correlations arise because the shared $\hat{\epsilon}$ is simultaneously stretched by multiple flow steps.

### B.3. Variance Reduction in FP++

For FP++, using independent perturbations, the covariance terms vanish:

$$\mathrm{Var}\left[\sum_i X_i\right] = \sum_i \mathrm{Var}[X_i] = \sum_i \mathrm{Var}[Y_i] < \mathrm{Var}\left[\sum_i Y_i\right]. \tag{33}$$

Thus, the total variance of the FP++ estimator is strictly smaller than that of FP. Intuitively, breaking the dependence between steps prevents the correlation between contributions from inflating the overall variance.

### B.4. Summary

- The key difference is that FP++ uses independent random directions per step, whereas FP reuses the same direction.

- The single-step variances are equal: $\mathrm{Var}[X_i] = \mathrm{Var}[Y_i]$.

- Covariance in FP arises from the shared perturbation being stretched similarly across flow steps.

- Removing these covariances in FP++ reduces the total variance, yielding a more efficient estimator.

## C. Gaussian Mixture Model Benchmark

In this work, we consider a Gaussian Mixture Model (GMM) to generate a multimodal test distribution. An $n$-dimensional GMM with $k$ components represents the probability density as

$$p(\mathbf{x}) = \sum_{j=1}^{k} \pi_j \, \mathcal{N}(\mathbf{x} \mid \boldsymbol{\mu}_j, \Sigma_j), \tag{34}$$

where $\pi_j$ is the mixture weight for the $j$-th Gaussian, and $\mathcal{N}(\mathbf{x} \mid \boldsymbol{\mu}_j, \Sigma_j)$ denotes the corresponding multivariate normal distribution.

For our experiments, we adopt a two-component GMM ($k = 2$) with the following setup:

- **Component Means ($\boldsymbol{\mu}_j$):** The first Gaussian is centered at $(-2, 0, \ldots, 0)$, and the second at $(2, 0, \ldots, 0)$, with the displacement applied only along the first dimension and all remaining dimensions set to zero.

- **Covariance Matrices** ($\Sigma_j$): The two components have distinct covariance structures: - Component 1 ($j = 1$) uses a diagonal covariance matrix with positive entries drawn as $0.01 + |\mathcal{N}(0, 0.25)|$, ensuring positive definiteness. - Component 2 ($j = 2$) employs a full covariance matrix: first, a diagonal matrix is sampled as above, then a random rotation (obtained via the QR decomposition of a random matrix) is applied to introduce correlations between dimensions.

- **Mixing Weights** ($\pi_j$): We set $\pi_0 = 0.25$ and $\pi_1 = 0.75$, creating an asymmetric mixture with the second component more heavily weighted.

The density can equivalently be expressed as a Boltzmann distribution,

$$p(\mathbf{x}) = \exp(-E(\mathbf{x})), \tag{35}$$

where $E(\mathbf{x})$ denotes the energy of configuration $\mathbf{x}$.

This particular GMM presents a challenging scenario for sampling, with two well-separated modes and heterogeneous covariance structures, making it a suitable benchmark for testing and validating advanced Boltzmann sampling techniques.

## D. Brute-Force and Stochastic Estimation of CNF Jacobian Traces

Continuous Normalizing Flows (CNFs) generalize standard Normalizing Flows by defining the mapping $\mathbf{f} : \mathbf{z} \to \mathbf{x}$ through an ordinary differential equation (ODE):

$$\frac{d\mathbf{x}_t}{dt} = \mathbf{v}(\mathbf{x}_t, t), \quad \mathbf{x}_0 = \mathbf{z}, \tag{36}$$

where $\mathbf{v}(\mathbf{x}, t)$ denotes the velocity field governing the transformation. The log-determinant of the flow Jacobian can then be expressed as an integral over the divergence of $\mathbf{v}$:

$$\log\left|\det \frac{\partial \mathbf{f}}{\partial \mathbf{z}}\right| = \int_0^1 \nabla \cdot \mathbf{v}(\mathbf{x}_t, t) \, dt, \tag{37}$$

with the divergence given by the trace of the Jacobian $\partial \mathbf{v} / \partial \mathbf{x}$.

We outline two standard approaches for computing this Jacobian trace: a direct (brute-force) method and a stochastic method using the Hutchinson estimator.

### D.1. Direct Jacobian Construction via Automatic Differentiation

Let $\mathbf{v} : \mathbb{R}^D \to \mathbb{R}^D$ be differentiable. A full Jacobian $\partial \mathbf{v} / \partial \mathbf{x}$ can be obtained using automatic differentiation frameworks (e.g., PyTorch):

1. For each output component $v_i(\mathbf{x})$, define the unit vector $\mathbf{e}_i \in \mathbb{R}^D$ with a 1 in the $i$-th position and zeros elsewhere, such that

$$v_i(\mathbf{x}) = \mathbf{e}_i^\top \mathbf{v}(\mathbf{x}).$$

2. Compute the gradient $\nabla_{\mathbf{x}} v_i(\mathbf{x})$ via backpropagation. This yields the $i$-th row of the Jacobian:

$$\nabla_{\mathbf{x}} v_i(\mathbf{x}) = \left[\frac{\partial v_i}{\partial x_1}, \frac{\partial v_i}{\partial x_2}, \ldots, \frac{\partial v_i}{\partial x_D}\right].$$

3. Stack all rows to assemble the full Jacobian:

$$\frac{\partial \mathbf{v}}{\partial \mathbf{x}} = \begin{bmatrix} \nabla_{\mathbf{x}} v_1(\mathbf{x}) \\ \nabla_{\mathbf{x}} v_2(\mathbf{x}) \\ \vdots \\ \nabla_{\mathbf{x}} v_D(\mathbf{x}) \end{bmatrix}.$$

In practice, PyTorch provides `jacrev` to efficiently compute this Jacobian.

### D.2. Hutchinson Estimator for Trace Approximation

The Hutchinson estimator offers a stochastic alternative for estimating $\text{Tr}(\partial \mathbf{v}/\partial \mathbf{x})$ without explicitly forming the full Jacobian:

1. Sample a random vector $\mathbf{u} \in \mathbb{R}^D$ from a standard normal or Rademacher distribution.

2. Compute the gradient of the scalar $\mathbf{u}^\top \mathbf{v}(\mathbf{x})$ w.r.t. $\mathbf{x}$:

$$\mathbf{g} = \nabla_{\mathbf{x}}(\mathbf{u}^\top \mathbf{v}(\mathbf{x})) = \frac{\partial(\mathbf{u}^\top \mathbf{v})}{\partial \mathbf{x}}.$$

3. Form the quadratic form $\mathbf{u}^\top \mathbf{g} = \mathbf{u}^\top (\partial \mathbf{v}/\partial \mathbf{x}) \, \mathbf{u}$, which is an unbiased estimator of the trace.

4. Repeat for $N$ independent random vectors $\mathbf{u}_i$ and average:

$$\text{Tr}\Big(\frac{\partial \mathbf{v}}{\partial \mathbf{x}}\Big) \approx \frac{1}{N} \sum_{i=1}^{N} \mathbf{u}_i^\top \frac{\partial \mathbf{v}}{\partial \mathbf{x}} \, \mathbf{u}_i.$$

Each sample requires one backpropagation pass. Averaging over multiple $\mathbf{u}_i$ reduces estimator variance, providing an efficient method for high-dimensional CNFs.

## E. Computational Cost of JVP/VJP-Based Estimators

### E.1. FP and FP++ Implementations: FD, JVP, and VJP

The inverse–Jacobian–vector product appearing in Eq. (10), $J_k^{-1}\epsilon_k = (\partial \mathbf{f}_k^{-1}/\partial \mathbf{x}_k)\, \epsilon_k$, can be implemented in three ways:

**Finite-difference (FD) approximation.** The product is approximated by perturbing the inverse flow:

$$J_k^{-1}\epsilon_k \approx \frac{\mathbf{f}_k^{-1}(\mathbf{x}_k + \delta\epsilon_k) - \mathbf{f}_k^{-1}(\mathbf{x}_k - \delta\epsilon_k)}{2\delta}.$$

**Forward-mode Jacobian-vector product (JVP).** The JVP computes the exact directional derivative along $\epsilon_k$:

$$\text{JVP}(\mathbf{f}_k^{-1}, \mathbf{x}_k, \epsilon_k) = \frac{d}{d\eta}\mathbf{f}_k^{-1}(\mathbf{x}_k + \eta\epsilon_k)\Big|_{\eta=0} = J_k^{-1}\epsilon_k.$$

**Reverse-mode vector-Jacobian product (VJP).** Given a cotangent $\epsilon_k$ at the output, the VJP computes

$$\text{VJP}(\mathbf{f}_k^{-1}, \mathbf{x}_k, \epsilon_k) = \epsilon_k^\top J_k^{-1}.$$

Although this produces a row vector, its squared norm is equal to that of the corresponding JVP output:

$$\|\epsilon_k^\top J_k^{-1}\|^2 = (\epsilon_k^\top J_k^{-1})(\epsilon_k^\top J_k^{-1})^\top = \epsilon_k^\top (J_k^{-1})^\top J_k^{-1} \, \epsilon_k = \|J_k^{-1}\epsilon_k\|^2.$$

**Delta entropy computation.** For all three implementations, the contribution to the flow-based entropy change at step $k$ is

$$\Delta S_k = -\frac{D}{2}\log\frac{\|J_k^{-1}\epsilon_k\|^2}{\|\epsilon_k\|^2},$$

### E.2. FD-based implementations in the main text

In the main text, FP and FP++ are implemented using *finite-difference* (FD) estimators. Here, Jacobian information is approximated by evaluating the inverse flow at perturbed inputs. This requires:

- 1 ODE pass for trajectory generation, and

- 2 additional ODE passes for entropy estimation,

yielding a total of 3 ODE passes per SMC step.

### E.3. JVP/VJP-based implementations

In this appendix and in Table 5, we report runtimes for implementations that use explicit vector–Jacobian (VJP) or Jacobian–vector (JVP) products. In this regime, entropy-related terms are computed via automatic differentiation without perturbing the inverse flow. As a result, FP and FP++ require:

- 1 ODE pass for trajectory propagation, and

- 1 ODE pass for the associated JVP/VJP computation,

yielding a total of 2 ODE passes per SMC step. Hutchinson-based estimators rely on multiple stochastic probes, resulting in a substantially higher number of ODE passes and longer wall-clock times.

### E.4. Wall-clock times and interpretation

Table 5 summarizes both the theoretical (ODE counts) and empirical (wall-clock) computational cost for different Jacobian estimators under the JVP/VJP regime. The columns **VJP Time** and **JVP Time** report measured wall-clock times for explicit vector–Jacobian and Jacobian–vector implementations, respectively. The results indicate that, for JVP/VJP-based implementations, the dominant computational cost arises from the number of ODE solver evaluations.

*Table 5.* Computational cost of different Jacobian estimators. Cost (# ODE passes) counts the number of ODE solver evaluations per SMC step, including trajectory propagation and entropy computation. VJP Time and JVP Time report the total wall-clock time incurred by implementations based on explicit vector–Jacobian or Jacobian–vector products, respectively.

| Method | Cost (# ODE passes) | VJP Time | JVP Time |
|---|---|---|---|
| Hutchinson G2 / R2 | 3 | 30.4 h | 48.9 h |
| Hutchinson G10 / R10 | 11 | 100 h | 176 h |
| FP (single-step) | 2 | 23.8 h | 30.8 h |
| FP++ (multi-step) | 2 | 23.8 h | 30.8 h |

## F. Molecular Dynamics Simulations of Chignolin Mutant

We performed molecular dynamics (MD) simulations to investigate the behavior of the Chignolin Mutant. A total of six independent trajectories were generated, each consisting of $10^7$ integration steps with a time step of 1 fs. All simulations were carried out at a temperature of 300 K, employing the CHARMM22 force field together with the implicit OBC2 solvent model. These conditions are consistent with those used for producing the training dataset.

Across all runs, the system remained confined within one of its metastable conformational states. Figure 6 illustrates two representative trajectories projected onto the chosen reaction coordinate (RC) plane.

### F.1. Key Hyperparameters for Reproducibility

Several hyperparameters are critical for reproducing the variance reduction observed in FP and FP++ methods. The exact hyperparameter values and full implementation details are available in our public code repository: :

> https://anonymous.4open.science/r/Flow_Perturbation_pp-9EC5

.

## G. Statement on the Use of Large Language Models

During the preparation of this manuscript, large language models (LLMs) were used in a limited manner solely for language editing purposes, such as improving clarity, grammar, and academic style. All aspects of the research conception, methodological development, experimental design, analysis of results, and the scientific conclusions presented in this paper were carried out independently by the authors.

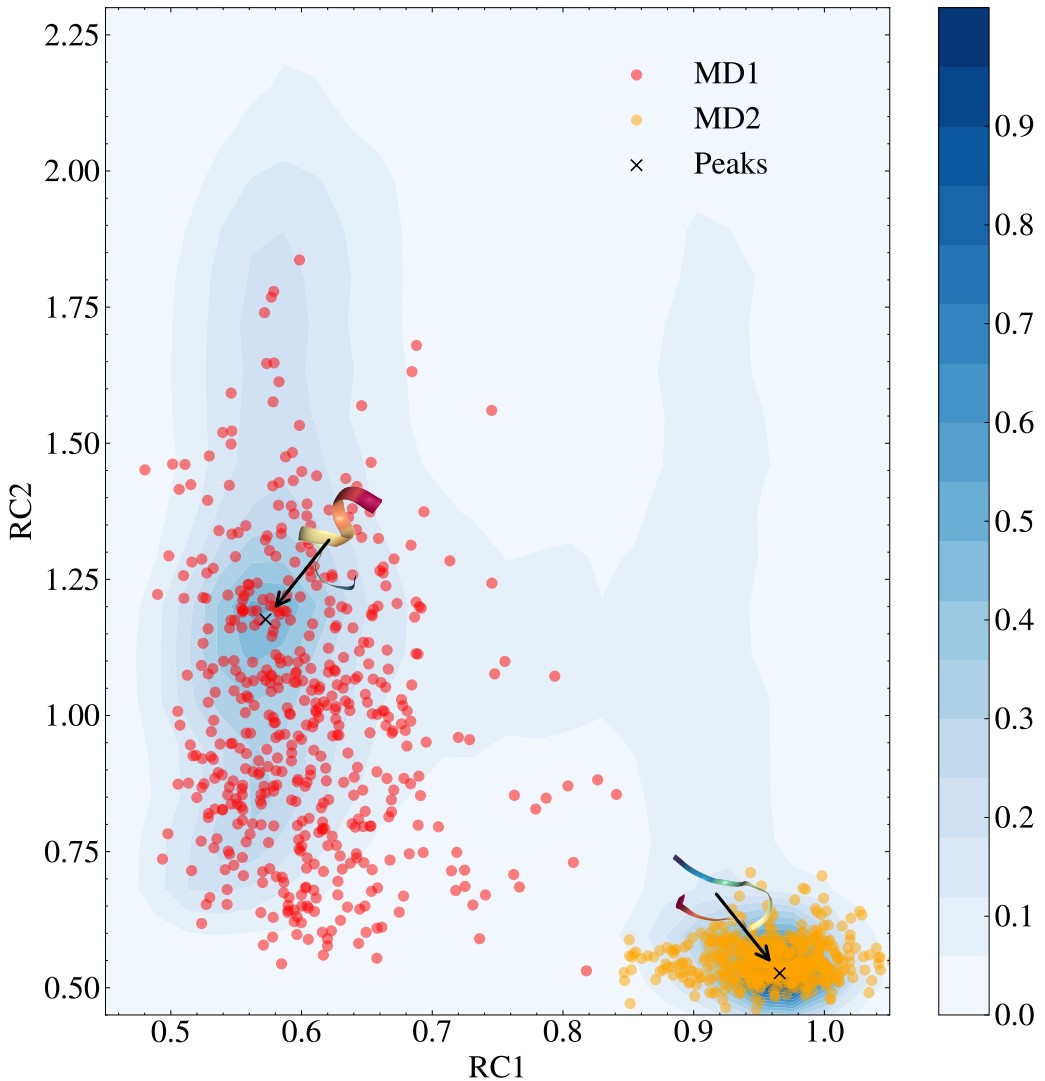

*Figure 6.* Representative MD trajectories of Chignolin Mutant visualized on the reaction coordinate plane.

