# OpenReview forum: "Flow Perturbation++: Multi-Step Unbiased Jacobian Estimation for High-Dimensional Boltzmann Sampling"
_ICML.cc/2026/Conference — Submitted to ICML 2026_

### Official Review · Reviewer_1dLQ · 2026-03-03

**Soundness:** 2
**Presentation:** 2
**Significance:** 2
**Originality:** 2
**Overall Recommendation:** 4
**Confidence:** 3

**Summary:**

# Summary

This paper introduces FP++, an improvement over the standard Flow Perturbation (FP) method for estimating the log-determinant of a flow's Jacobian. Instead of applying a single noise realization across the entire flow trajectory, FP++ injects multiple independent noises at each step. This modification is highly intuitive and comes with theoretical guarantees of reduced variance. Empirically, the sampling benchmarks demonstrate that FP++ effectively improves upon the original FP method and achieves competitive accuracy when compared to Hutchinson-based estimators.

However, there are several significant concerns that limit the paper's overall impact. Primarily, the novelty is somewhat incremental, as the core mechanism relies heavily on the existing FP framework. Furthermore, there are critical issues regarding the claims of unbiasedness, confounding factors in the experimental design, and the clarity of the presentation, which are detailed in the sections below.

**Compliance With Llm Reviewing Policy:**

Affirmed.

**Final Justification:**

I thank the author for providing additional experiments that solve most of my concerns. I will increase the score from a weak-reject to a weak-accept but lower my confidence.

**Key Questions For Authors:**

# Questions for the Authors

The questions are stated in the previous sections. Here are the recaps for clarity

**1. Finite-Difference Bias vs. Theoretical Unbiasedness:**
The paper emphasizes that FP++ is an unbiased estimator, yet the practical implementation relies on finite-difference approximations, which inherently introduce bias.
* *Question:* Can you provide a rigorous analysis isolating the bias introduced by the finite-difference approximation? Specifically, could you report experimental results using inverse-mode JVP (without finite-difference) to demonstrate the true unbiased performance of the estimator?
* *Impact on Evaluation:* Addressing this contradiction is critical for the paper's soundness. Demonstrating the isolated impact of this approximation error will validate your theoretical claims and significantly improve my score for technical rigor.

**2. Intrinsic Evaluation on GMM Benchmarks:**
Evaluating the estimators using downstream SMC sampling on GMM benchmarks introduces confounding variables (e.g., the number of particles). Since everything in GMMs is analytical, the optimal velocity is directly available.
* *Question:* Could you provide an intrinsic evaluation that directly compares the exact log-probability of the target GMM at a generated point $x$, using the exact velocity/score, against the estimated log-probability of the flow at $x$, without using the SMC loop?
* *Impact on Evaluation:* Providing this direct comparison will remove the noise introduced by SMC and prove the intrinsic accuracy of your estimator, which is essential for validating the empirical soundness of the paper.

**3. Fair Baselines for Accuracy (Table 3):**
In Table 3, the `BFJacobian` baseline performs very poorly, which seems more attributable to the compounding errors of the SMC setup rather than a failure of the exact calculation.
* *Question:* To measure true accuracy, can you compare the difference between your estimations and the exact `BFJacobian` *without* incorporating the SMC loop?
* *Impact on Evaluation:* A direct comparison to the exact `BFJacobian` will provide a much fairer baseline for accuracy. If FP++ proves competitive in this unconfounded setting, it will strongly solidify your empirical claims.

**4. Broader Significance and Alternative Paradigms:**
Estimating the divergence is not the only way to enable Boltzmann sampling. Recent methods [1, 2] achieve this using only the velocity or the score by estimating path measures or the joint likelihood of a denoising sequence, bypassing divergence estimation entirely.
* *Question:* How does your approach compare to these velocity/score-based methods? Can you provide a discussion (and ideally, an empirical comparison) justifying why estimating the divergence via FP++ remains necessary or advantageous in this context?
* *Impact on Evaluation:* Properly positioning your work against these alternative, concurrent paradigms will greatly enhance the "Significance" and "Originality" dimensions of my review.

---

**References**

[1] Skreta, Marta, et al. "Feynman-Kac correctors in diffusion: Annealing, guidance, and product of experts." arXiv preprint arXiv:2503.02819 (2025).

[2] He, Jiajun, et al. "RNE: a plug-and-play framework for diffusion density estimation and inference-time control." arXiv e-prints (2025): arXiv-2506.

**Limitations:**

The limitations are not discussed. While they are obvious based on the above sections: the practical FP++ is biased, unlike what the author claimed, and therefore requires systematic analysis.

**Strengths And Weaknesses:**

# Soundness

While the proposed FP++ method is theoretically sound and successfully demonstrates a variance reduction compared to the original FP estimator, there are several critical gaps between the theoretical claims and the empirical evaluation that undermine the paper's overall soundness:

1. **Theory vs. Practice Gap (Unbiasedness):** The authors repeatedly claim that both the FP and FP++ estimators are unbiased. However, in practice, the method relies on finite-difference approximations, which inherently introduce bias. Claiming unbiasedness while deploying a biased practical implementation is contradictory. The authors must rigorously analyze the bias introduced by the finite-difference approximation. Alternatively, they should provide experimental results without finite-difference (e.g., using inverse-mode JVP) to isolate and justify this approximation error.
2. **Confounded Experimental Evaluation:** Evaluating the estimator using SMC-generated samples on GMM benchmarks is suboptimal and introduces confounding variables. Since GMMs are analytical, the optimal velocity (the "model") is available directly. Therefore, given $x$ generated by the flow, the intrinsic and rigorous comparison should be the exact log-probability of the target GMM at $x$ versus the estimated log-probability of the flow at $x$.
3. **Unfair Baselines in Table 3:** Evaluating accuracy via SMC introduces unnecessary noise. SMC generally requires many particles to achieve good performance. In Table 3, `BFJacobian` performs very poorly, due to the SMC setup. A more direct analysis comparing different estimations against the exact `BFJacobian` (without the SMC loop) would provide a much clearer assessment of accuracy, even if it requires a higher computational budget for the evaluation phase. In fact, there are two ways:
    1. Since everything in GMMs are analytical, the optimal velocity, which could be treated as the “model”, is available directly. In this case, the model density is just the log-prob of the target GMM. And therefore, given x generated by the flow, directly comparing the “true log-prob of GMM at x” and the “estimated log-prob of flow at x” is much more ideal. Furthermore, the exact GMM setting also allows closed-form and analytical access to the marginal log densities, and hence enable timestep-wise analysis.

    2. Alternatively, the author can also consider comparing the difference between different estimations with the “BFJacobian”, which can also avoid doing the SMC. This allows analysis beyond the optimal/perfect velocity, but brings much more computational budget for sure.

# Presentation

The paper is generally understandable, but several structural and notational issues need to be addressed to improve clarity and avoid misleading the reader:

1. **Inconsistent Notation:** The mathematical notation is frequently unclear. For example, $f$ is used as the drift of the SDE (L120), while bolded $\mathbf{f}$ denotes the flow (L134). The notation for the Jacobian is also inconsistent; an input is specified in L150 ($\partial v(x)/\partial x$) but omitted in L160. Furthermore, $T$ is used for the number of flows on the right column of L117, but this switches to $K$ in L128. These inconsistencies make the technical narrative difficult to follow.
2. **Overstated Claims Regarding SMC:** In L91-93, the authors state: *"FP++ integrates naturally with SMC annealing, providing unbiased and variance-reduced Boltzmann reweighting in high dimensions."* This phrasing misleadingly implies that the natural integration with SMC is a novel strength unique to FP++. In reality, standard baselines can also be naturally integrated with SMC. This claim should be softened.
3. **Clarity in Table 1:** The computational overhead presented in Table 1 is slightly confusing. For FP/FP++ without finite-difference, one can use the same $N$ probes as Hutchinson in principle, resulting in 1+$N$ passes. If using finite-difference, the overhead would be 2$N$+1. The authors should clearly delineate this and emphasize that FP/FP++ with finite-difference and $N$=1 is already accurate, resulting in 3 passes in practice.

# Significance

Estimating the model density (log-likelihood) of continuous flows is an important and highly relevant problem in generative modeling. Accurate estimations unlock significant potential for downstream tasks like Boltzmann sampling, reward-tilting, and model-composition, where both of these tasks could be employed SMC or other advanced sampling methods that ultilze the log-(marginal-)density. The main contribution—trading bias for efficiency by avoiding expensive JVPs via FP++ with finite-difference—provides practical utility.

However, the significance of this specific approach is limited by the omission of a broader literature context. Estimating the divergence is not the only way to enable Boltzmann sampling, e.g. with SMC, from trained diffusion or continuous-flow models. Recent methods only require the velocity or the score to enable SMC by estimating path measures or the joint likelihood of a denoising sequence [1, 2]. To truly demonstrate the significance of FP++, the authors need to discuss and empirically compare against these alternative paradigms to justify why estimating the divergence remains the superior or necessary path.

# Originality

The theoretical originality of the paper is clear, as FP++ provides a smart, variance-reduced improvement over the standard FP estimator without imposing additional overhead. However, the empirical originality is somewhat limited. Despite the theoretical advancements, the experimental setups and benchmarks are almost identical to those in the original FP paper, which makes the contribution feel somewhat incremental in its current state. Expanding the scope of the experiments (e.g., applying it to novel tasks or directly comparing to the velocity-based correctors mentioned above) would greatly enhance the work's originality.

---

**References**

[1] Skreta, Marta, et al. "Feynman-Kac correctors in diffusion: Annealing, guidance, and product of experts." arXiv preprint arXiv:2503.02819 (2025).

[2] He, Jiajun, et al. "RNE: a plug-and-play framework for diffusion density estimation and inference-time control." arXiv e-prints (2025): arXiv-2506.

---

> ### Author Rebuttal · Authors · 2026-03-31
>
> We thank the reviewer for the constructive comments. We have provided clarifications and additional experimental analyses regarding computational cost, estimator variance and convergence, finite-difference bias, comparison to related work, and notation consistency.
>
> We emphasize that these clarifications and additional results, provided in the anonymous repository
> \url{https://anonymous.4open.science/r/Reviewer2-F41B},
> further strengthen the empirical and theoretical validity of the paper, and do not change the main conclusions.
>
> **1. Finite-Difference Bias and Theoretical Consistency**
>
> In practice, finite differences introduce truncation error controlled by the perturbation scale $\delta$.
>
> In our experiments, we use $\delta = 0.001$. To quantify the effect of this choice, we compare the finite-difference FP++ estimator with the JVP-based implementation under identical evaluation protocols.
>
> Specifically, we conduct $M$ independent runs with fixed computational budgets to evaluate the estimator error.  The estimation error is defined as the average absolute difference between the estimator and the true absolute Jacobian determinant $\left| \det \left( \frac{\partial f}{\partial z} \right) \right|$ over all runs and samples.
>
> The full results in the anonymous repository show that:
>
> (1). Both implementations closely match the ground-truth Jacobian determinant;
>
> (2). Their error curves almost completely overlap across iterations;
>
> These observations indicate that the finite-difference bias is negligible in the considered experimental setting.
>
> **2. GMM Design, SMC Effects, and Intrinsic Estimation Error**
>
> We refer the reviewer to our response to Reviewer X5dr on *2. Estimator Variance and Convergence Behavior*, where we provide a detailed analysis of variance reduction, convergence dynamics, and intrinsic estimator error under controlled experimental settings. The same experimental evidence applies to the present concern.
>
> **3. Relation to FKC / RNE and Conceptual Differences**
>
> We thank the reviewer for pointing out related works such as Feynman-Kac Correctors (FKC) and Radon-Nikodym Estimators (RNE). However, FP++ differs fundamentally in both objective and statistical interpretation.
>
> (1). **Target distribution and objective difference:**
>
> FKC and RNE rely on classifier-free guidance (CFG)-style ideas within an SMC framework, and do not perform Boltzmann sampling. In particular, they do not explicitly evaluate the energy function during the process. As a result, they do not recover the Boltzmann distribution, but rather remain tied to the data distribution learned by the pretrained model.
>
> Even with reward-tilting, the resulting distribution is still defined as
> $
> p(x) \propto q_{data}(x) \exp(r(x)),
> $
> which fundamentally depends on the original dataset distribution and its biases.
>
> (2). **Mode correction and distributional bias:**
>
> In the GMM 10D experiment (training 50/50 vs target 25/75), we evaluate FKC-style methods and observe that they fail to correct the inherited mode imbalance, while FP++ successfully recovers the target proportions.
>
> This behavior arises because FKC/RNE-style methods do not compute absolute density ratios, and therefore remain tied to the bias of the proposal distribution. The corresponding results are provided in the anonymous repository.
>
> In contrast, FP++ is able to correct the induced mode imbalance through the Jacobian determinant term, which directly accounts for volume changes in the transformation, leading to improved recovery of the correct mode proportions independent of the proposal distribution.
>
> **4. Notation and Complexity Clarification**
>
> We unify notation and complexity descriptions in the revised manuscript:
>
> (1). Drift field is denoted as $b(x_t, t)$ and flow map as $\mathbf{f}$;
>
> (2). We replace $\frac{\partial v(x)}{\partial x}$ in L150 with $\frac{\partial v}{\partial x}$, omitting explicit input variables for consistency throughout the manuscript.
>
> (3). We replace $K$ in L117 with $T$ for consistency throughout the manuscript.
>
> (4). Clarify in Table 1 that the computational cost of FP/FP++ under finite-difference is $1 + 2N$ passes; in particular, for $N=1$, this corresponds to 3 passes in our experiments.

---

> > ### Author Rebuttal · Reviewer_1dLQ · 2026-04-01
> >
> > Thank you for the clarification. While the anonymous link seems to be broken, which is not able to be opened.
> >
> > The following concerns are not solved and remain, even though I assume the additional experiments (which I can't see now) are the same as claimed by the authors:
> >
> > 1. Theory vs. Practice Gap (Unbiasedness)
> >
> > the additional experiments are claimed that this introduces negligible error, however, it is unclear that how this behaves between different systems. It might be fragile on complex ones. Hence more justification should be made here.
> >
> > 2. Regarding "Relation to FKC / RNE and Conceptual Differences"
> >
> > RNE is able to do Boltzmann sampling, which is provided in their C.1 and C.2 with experiments in F.
> >
> > 3. Regarding GMM experiments
> >
> > Since GMM is a highly analytical distribution, smarter evaluations should be considered. In particular, given a noising process, all the intermediate (marginal) distributions are GMMs, where the velocity and its Jacobian are analytical. In such a case, a more informative comparison is to see if the estimator could be accurate under the perfect velocity. This would not suffer from the computational issue of the BF one as stated in the tables.
> >
> > Since the paper is proposing an estimator that claimed to be unbiased and accurate, a more systematic evaluation that removes the proxy, i.e. SMC, is more desired.
> >
> > ---
> >
> > Overall, I recommend the author to fix the anonymous link first. Based on the broken link and unsolved concerns, I will maintain my score.

---

> > > ### Author Response · Authors · 2026-04-02
> > >
> > > We thank the reviewer for the careful and constructive feedback. We first note that the previously provided anonymous link may have been broken due to a formatting issue (the trailing punctuation “\}.” was incorrectly parsed into the URL). A corrected and fully accessible repository is now available at:
> > >
> > > https://anonymous.4open.science/r/Reviewer2-F41B
> > >
> > > We provide detailed point-by-point responses below.
> > >
> > > ***1. Theory vs. Practice Gap (Unbiasedness and δ-sensitivity)***
> > >
> > > We appreciate the concern regarding the unbiasedness of the finite-difference FP++ estimator and its $\delta$ sensitivity across different systems and dimensions.
> > >
> > > To address this, we conducted a systematic $\delta$-sensitivity study across multiple dimensional settings, comparing the finite-difference FP++ estimator with the JVP-based implementation under identical evaluation protocols.
> > >
> > > We vary $\delta$ over a wide range and observe the following:
> > >
> > > - In 10D, $\delta = 0.1$ yields nearly overlapping results between finite-difference and JVP estimators, indicating negligible discretization error.
> > >
> > > - In 100D and 1000D settings, stable agreement is observed at $\delta = 0.05$.
> > >
> > > To further address the reviewer’s concern, we additionally evaluate a step size $\delta = 0.001$ on the Chignolin mutant system, where finite-difference estimates still closely match those of the JVP estimator.
> > >
> > > The value used in the paper ($\delta = 0.001$) is therefore well within the stable regime.
> > >
> > > These results indicate that the finite-difference approximation operates in a stable regime for sufficiently small $\delta$, and does not exhibit noticeable degradation across the tested dimensional settings. Moreover, as observed in our convergence experiments (see discussion below), with $\delta = 0.001$, the FP++ estimator converges more accurately to the Jacobian determinant than the Hutchinson estimator, consistently across 400 independent runs ($M=400$) on GMM 10D, GMM 100D, and GMM 1000D.
> > >
> > > ***2. Relation to RNE / FKC and Boltzmann interpretation***
> > >
> > > We thank the reviewer for raising this point.
> > >
> > > We first clarify the objective of RNE by quoting the original formulation from the caption of Fig. 11 in the cited paper:
> > >
> > > "where $t \in [0, 10],\; p_{t=10} \rightarrow \mathcal{N}(0, 10^2 I)$ and $p_0 = p_{data}$."
> > >
> > > This explicitly states that the terminal objective is $p_0 = p_{data}$, i.e., recovering the data distribution.
> > >
> > > Therefore, RNE is fundamentally designed for distribution matching to $p_{data}$, and any Boltzmann connection only applies in the special case where the data distribution itself satisfies $p_{data} \propto \exp(-E(x))$.
> > >
> > > To further examine the behavior when the data distribution differs from the target Boltzmann distribution, we evaluate RNE on an imbalanced Gaussian Mixture Model, where the training distribution is 50/50 while the target Boltzmann distribution is 25/75. The method converges to an approximately 50/50 marginal distribution, rather than achieving the ground-truth imbalance, as shown in the anonymous repository.
> > >
> > > This empirical result is consistent with the interpretation that RNE matches the induced $p_{data}$ rather than enforcing a general Boltzmann weighting.
> > >
> > > ***3. Accuracy evaluation and Jacobian determinant convergence***
> > >
> > > We thank the reviewer for this insightful suggestion regarding the evaluation methodology.
> > >
> > > We would like to clarify that our previous rebuttal experiments already tried to remove the influence of SMC. In the anonymous repository, we directly compare the proposed estimator with the ground-truth Jacobian determinant, without relying on any SMC-based approximation or proxy. The experimental results show that FP++ matches the ground-truth Jacobian determinant very closely. Therefore, the reported results reflect the intrinsic accuracy of the estimator itself, rather than variability introduced by SMC sampling.
> > >
> > > To further address the reviewer’s concern and provide a more informative evaluation under fully analytical settings, we additionally construct a 1000D GMM where both the velocity field and its Jacobian are available in closed form. In this setting, we can compute the exact Jacobian determinant along the entire noising trajectory.
> > >
> > > Under this fully analytical benchmark, we show that FP++ provides highly accurate estimates of the Jacobian determinant across the entire time interval. Specifically, we randomly sample initial points and report the Log-Jacobian determinant over time in $t \in [0,1]$, where at each time step we compute the Log-Jacobian determinant for multiple sampled points and then take their average. This demonstrates close agreement with the analytical ground truth. The results are provided in the anonymous repository.
> > >
> > > We will clarify this point and add the corresponding description in the revised manuscript.

---

### Official Review · Reviewer_X5dr · 2026-03-15

**Soundness:** 3
**Presentation:** 3
**Significance:** 2
**Originality:** 2
**Overall Recommendation:** 4
**Confidence:** 3

**Summary:**

In this work the authors present an extension on the recently introduced Flow Perturbation (FP) method for Jacobian estimation in Continuous Normalizing Flows (CNFs). The primary motivation of the FP method is that full Jacobian evaluation is too expensive for large systems, and that the Hutchinson approximation for this can be biased. As such, the authors of the FP method instead propose to use a Jacobian determinant estimation using a probe vector along which the gradient is calculated. In practice, this is implemented using a finite difference approximation rather than exact automatic differentiation.

In this work, the authors extend on this by not only doing this probe at the end of the generation, but also for intermediate steps. They argue that this reduces the variance of the estimator, which is formally proven in Appendix B. The authors evaluate their method on the same set of problems as used for FP: a low-dimensional (10D) and high-dimensional (1000D) GMM, and the Chignolin system.

**Compliance With Llm Reviewing Policy:**

Affirmed.

**Final Justification:**

I thank the authors for their respond and addressing my questions and concerns. The authors have addressed all my concerns experimentally.

However, I remain doubtfull that with a single probe the full global volume information can be uncovered. While unbiased I would still expect high variance in the estimator. The newly presented results however suggest otherwise.

As such, I will increase my score to a weak accept, but will drop my confidence.

**Key Questions For Authors:**

Can you clarify the number of probe samples N used in all experiments, and provide a study of how estimator variance and accuracy scale with N across different dimensionalities?

**Limitations:**

Yes

**Strengths And Weaknesses:**

The authors present a well-written paper that discusses an important problem in the use of CNFs for tasks where density estimation is required, in this case unbiased Boltzmann sampling. The paper has sufficient coverage of the background to be easy to follow.

However, the paper suffers from some significant weaknesses that are mostly inherited from the original FP method. Unfortunately these initial weaknesses are insufficiently addressed by the authors of this follow-up paper to warrant acceptance in its current form. These issues primarily center around (1) insufficient coverage of the method's possible high variance and (2) concerns with the experimental evaluation. I address each below.

**Possible High Variance**

The primary benefit the authors claim FP has over the Hutchinson method is that FP is unbiased while Hutchinson is not. While this is true, FP remains an estimator and not an exact method. As such, to justify its use, a study of the estimator's variance and general convergence is needed. This is not provided in the original paper that introduces FP, and is also omitted here.

Based on the text, it appears to be the case that for all experiments only a single probe is used, but this is never explicitly mentioned. This seems very unlikely in high dimensions, where a single unit vector probe would capture only very limited information about the Jacobian. This should be clarified by the authors.

The issue of possible high variance also shows up in the analysis of the methods cost comparison. The paper reports the Hutchinson cost to be 1+N, but reports a fixed cost of 3 for FP and FP++, again implying that it is always sufficient to take only a single probe. This is, in my opinion, very unlikely and therefore feels like a misleading representation of the true cost. Discussing the costs as being 1+2N and including an analysis of the number of N, and how it is impacted by dimensionality, would be appropriate.

**Experimental evaluation raises concern**

The experimental evaluation used to analyze FP++ and compare it against FP and the Hutchinson estimator is borrowed from the FP paper. While it makes sense to use the same evaluation, I unfortunately believe that this original setup does not sufficiently cover possible issues with the method. This is most notable in the choice of primarily using GMMs for the evaluation.

Using a normalized Gaussian prior and, what seems to be a normalized, GMM target distribution implies that we have a Jacobian close to one. This should naturally suppress estimator variance by making all dimensions have roughly the same JVP, and as such would explain why both FP++ and FP seem to only require a single sample N=1 to accurately recover the density ratios. It is unlikely that this will generalise to more complex systems.

The one system on which this concern is most relevant is the Chignolin system, for which the experimental setup is unfortunately not reported clearly enough to determine whether N=1 was used. In addition, the paper does not report any error bars for this experiment, as only a single SMC run was performed. Lastly, the FP results presented here (0.475) do not match those reported in the original FP paper (0.433 for FP1), which is noteworthy even accounting for minor differences in experimental setup such as the number of annealing steps.

**Minor issues:**

Lastly, two minor issues that would be good to get rectified in future versions of the paper:
1. Eq. 1: the left-hand side should have E_{x~p}[O(x)]
2. Line 114-116: The unbiasedness claim applies to the determinant not the entropy/log-determinant. The normalized importance weights used in practice make the estimator consistent, not unbiased.

---

> ### Author Rebuttal · Authors · 2026-03-31
>
> We thank the reviewer for the suggestion and have conducted additional experiments to address the concerns. All additional figures and extended experimental results are provided in the anonymous repository:
> \url{https://anonymous.4open.science/r/Reviewer-73D4}.
>
> We address the comments point by point below.
>
> **1. Computational Cost and Number of Probes $N$**
>
> We agree that FP++ is not restricted to the single-probe setting. In general, its computational cost is
> $1 + 2N$,
> where $N$ is the number of probes.
>
> In the manuscript, we reported a cost of $3$ because all experiments were conducted under $N=1$, corresponding to $1 + 2 \times 1$. We will clarify this explicitly in the revision.
>
> Importantly, this experimental setting does not affect our conclusions. While the original manuscript only reports results for $N=1$, we additionally evaluate FP++ across different values of $N$ in the extended experiments as shown in the anonymous repository. The results show that FP++ consistently improves over FP, and the main performance gain comes from the variance reduction introduced by FP++.
>
> **2. Estimator Variance and Convergence Behavior**
>
> We agree that systematic variance analysis is essential. We therefore provide extensive empirical evaluations under identical computational budgets in the anonymous repository.
>
> **(a) Variance across dimensionality and number of probes $N$.**
> We report entropy estimator variance computed using 1000 samples for GMMs in 10D, 100D, and 1000D. The results show:
>
> (1). FP++ consistently achieves significantly lower variance than FP across all dimensionalities;
>
> (2). FP++ achieves variance levels that are **comparable to the Hutchinson estimator**;
>
> (3). Increasing $N$ leads to only marginal improvements, suggesting diminishing returns from probe averaging.
>
> These results are provided in Table 1 of the anonymous repository.
>
> **(b) Convergence under fixed computational budgets.**
> We further conduct $M$ independent runs with fixed number of samples to evaluate the estimator error. The estimation error is defined as the average absolute difference between the estimator and the true absolute Jacobian determinant $\left| \det \left( \frac{\partial f}{\partial z} \right) \right|$ over all runs and samples.
>
> The results show:
>
> (1). FP++ variants rapidly converge within approximately $M=70$, as shown in the convergence curves;
>
> (2). FP++ consistently achieves lower estimation error than Hutchinson under matched cost;
>
> (3). Increasing $N$ reduces the $M$ required to reach convergence, but the improvement quickly saturates.
>
> Overall, FP++ provides a strong trade-off between computational efficiency and estimation accuracy, reducing variance compared to FP while preserving its unbiasedness. Under the same computational budget, FP++ also achieves more accurate estimates than the Hutchinson estimator.
>
> **3. Validity of the GMM Benchmark**
>
> We clarify that the 1000D GMM used in our experiments is non-trivial, featuring strong anisotropy induced by structured covariance matrices. One component uses a diagonal covariance with heterogeneous variances, while the other applies a random orthogonal rotation to a diagonal covariance, introducing strong cross-dimensional correlations. As a result, the model is far from an identity-Jacobian setting.
>
> If the Jacobian were close to identity, the Hutchinson estimator would be expected to perform competitively. However, we observe significant degradation of Hutchinson performance in high dimensions, while FP++ remains consistent. This indicates that the benchmark contains substantial geometric and spectral complexity, validating the experimental setting.
>
> **4. Stability on the Chignolin System**
>
> To further evaluate the stability of the FP++ estimator, we conduct 10 independent runs on the Chignolin system. We report the mean and standard deviation across runs in Table 2, while the full results are provided in the anonymous repository.
>
> Table 2: Chignolin (mean ± std) over 10 runs
>
> | Method | Metric      | Mean   | Std   |
> |--------|-------------|-------:|------:|
> | FP++   | Proportion  | 0.3658 | 0.0146 |
> | FP++   | # Preserved | 1473.8 | 18.0   |
>
> The results show:
>
> (1). FP++ exhibits consistent performance across independent runs;
>
> (2). Minor discrepancies in FP results arise from stochasticity in SMC sampling and implementation-level variability;
>
> Such variations do not affect the relative performance ordering of methods.
>
> **5. Correction of Mathematical Formulation**
>
> We thank the reviewer for pointing out the ambiguity. In the revision, we will:
>
> (1). Correct Equation (1) to $\mathbb{E}_{x \sim p}[O(x)]$;
>
> (2). Clarify that the unbiasedness property applies to the determinant estimator $\hat{J}_{\mathrm{FP++}}$, i.e.,
>
> $\mathbb{E}[\hat{J}_{\mathrm{FP++}}] = \left| \det \frac{\partial f}{\partial z} \right|$.
>
> Our convergence experiments further validate the unbiased behavior of the determinant estimator.

---

> > ### Author Rebuttal · Reviewer_X5dr · 2026-04-01
> >
> > I thank the authors for their response and addressing my questions and concerns. The authors have addressed all my concerns experimentally.
> >
> > However, I remain doubtfull that with a single probe the full global volume information can be uncovered. While unbiased I would still expect high variance in the estimator. The newly presented results however suggest otherwise.
> >
> > As such, I will increase my score to a weak accept, but will drop my confidence.

---

### Official Review · Reviewer_AqpS · 2026-03-17

**Soundness:** 3
**Presentation:** 3
**Significance:** 3
**Originality:** 3
**Overall Recommendation:** 4
**Confidence:** 4

**Summary:**

This paper introduces a new method for unbiased estimation of the reweighting factor in flow-based models for sampling from Boltzmann distributions. The method serves as a drop-in replacement for the standard Hutchinson estimator with comparable or improved computational efficiency, but with the advantage of being unbiased. It builds on an earlier method called flow perturbation that also provides unbiased estimates, but decomposes the computation into substeps to reduce variance, which the authors show improves results on standard benchmarks. The method is simple, the paper is overall well-written, and the empirical results are impressive.

**Compliance With Llm Reviewing Policy:**

Affirmed.

**Final Justification:**

My overall take on the paper is essentially that of my initial review. I believe the authors tackle an important (though not groundbreaking) problem and bring some solid technical machinery to the table. I am positive, though not overwhelmingly so, about the manuscript.

**Key Questions For Authors:**

1. The paper claims FP++ requires only three passes, but with $L$ substeps shouldn't the cost be $3L$? What exactly is being counted? Maybe I am misunderstanding something about the method. Can the authors clarify this?

2. How is $f_k^{-1}$ obtained in practice? Is it simply backward integration of the flow?

3. Under what conditions does weight degeneracy in the SMC reweighting become a problem? Have you observed concentration of importance weights on a small number of samples in the high-dimensional experiments?

4. Can you clarify the scope of applicability? The Boltzmann framing is very general, but the method requires a trained flow model, which in turn requires samples or a tractable energy function. In what practical settings beyond molecular simulation does this apply?

**Limitations:**

The authors should discuss the scaling of computational cost with the number of substeps $L$ more clearly. The potential for weight degeneracy in high dimensions deserves more attention. The scope of applicability claims in the introduction should be tempered.

**Strengths And Weaknesses:**

*Soundness:*

The core contribution - an unbiased, low-variance estimator of the reweighting factor that serves as a drop-in replacement for Hutchinson - is technically clean and well-supported by experiments. The multi-step decomposition is a natural idea and the variance reduction it provides is convincingly demonstrated. However, I do have some concerns. The authors claim FP++ requires only three passes, but if the method uses $L$ substeps, naively I would expect the cost to scale as $3L$ since each substep requires its own passes; this needs clarification. Computing the inverse Jacobian-vector product requires access to $f_k^{-1}$, but the paper does not explain how this inverse is obtained (e.g., is it just backward integration of the flow?). Additionally, importance weights can concentrate on a few samples in high dimensions, and even with SMC reweighting this could simply result in duplicating those samples; the authors should comment on when weight degeneracy is or is not a concern.

*Presentation:*

The paper is overall well-written and the method is clearly presented, but there are several issues. The discussion of the original flow perturbation method lacks sufficient detail for self-containedness: the notation ($\|df^{-1}/dx\|$, the norm, the ${}^{-D}$ exponent) is unclear for readers unfamiliar with the original paper. $f$ is overloaded as both the SDE drift and the flow map. The explanation of why the Hutchinson estimator is biased should be more precise: it gives an unbiased estimate of the log probability, but the exponential needed for reweighting introduces a nonlinearity that makes it biased. The flow matching literature is cited only via Lipman et al. and should include the rectified flow (Liu et al.) and stochastic interpolants (Albergo, Boffi, and Vanden-Eijnden) papers. The figures (e.g., Figure 3) use colors that are hard to distinguish, and the tables should bold the best results.

*Significance:*

Unbiased Boltzmann sampling is an important problem and the method addresses a real practical limitation of existing approaches. The introduction overstates the scope somewhat by claiming applicability to settings like RL where one may not have the samples needed to train the flow model in the first place. Any distribution can be written as a Boltzmann distribution, so this framing is vacuous; the authors should be more precise about actual applicability.

*Originality:*

The multi-step decomposition of flow perturbation for variance reduction is a clean and natural extension of prior work. The contribution is incremental but practically useful.

---

> ### Author Rebuttal · Authors · 2026-03-31
>
> We thank the reviewer for the positive assessment and for pointing out the need to clarify the computational cost, implementation details, scope of applicability, and notation. We address each point below.
>
> **1. Computational Cost: 3 Passes vs. $3L$**
>
> We define the computational cost in terms of "passes," where one pass is equivalent to a full ODE integration across all steps. To clarify how FP++ incurs a total of three passes, we decompose the process into its trajectory evaluation and Jacobian estimation two parts:
>
> (1). A forward ODE integration generates the sample trajectory $x = f(z)$ and stores intermediate states $\\{x_k\\}_{k=1}^L$;
>
> (2). At each step, FP++ evaluates the inverse map $f_k^{-1}$ at two perturbed inputs to compute a finite-difference estimate of the Jacobian at that step. Although these evaluations are performed at each of the $L$ steps, they do not require recomputing the full ODE trajectory. When aggregated over all steps, their total cost is comparable to two full ODE integrations.
>
> Therefore, the total cost remains
> $
> 1\ (\text{generation}) + 2\ (\text{Jacobian estimation}) = 3\ \text{passes},
> $
> and **does not scale linearly with $L$**. We will clarify this more explicitly in the revision.
>
> **2. Implementation of $f_k^{-1}$**
>
> In Continuous Normalizing Flows (CNFs), the flow is defined by the ODE
> $
> \frac{dx}{dt} = v(x,t).
> $
> For a discrete step of size $\Delta t$, the local inverse map $f_k^{-1}$ is obtained by **integrating the ODE backward**, i.e., from $t+\Delta t$ to $t$.
>
> We will add this clarification in the revised paper.
>
> **3. Weight Degeneracy in High Dimensions**
>
> We had already conducted experiments demonstrating that weight degeneracy is a fundamental challenge in high-dimensional importance sampling. To quantify this effect, we use the **Remaining Ancestor Count**, which measures the number of surviving trajectories that survive resampling. A lower ancestor count indicates more severe weight collapse, as only a small fraction of particles contribute to the final population.
>
> In the 1000D GMM experiment:
>
> (1). FP exhibits severe degeneracy, with only **59** surviving ancestors;
>
> (2). FP++ significantly alleviates this issue, maintaining **312** ancestors.
>
> This improvement arises because FP++ reduces the variance of the weight estimator, leading to more evenly distributed importance weights. As a result, it improves the robustness of the SMC procedure in high dimensions.
>
> To directly evaluate the intrinsic behavior of the estimator, we additionally provide a variance and convergence analysis of FP++ without the influence of the SMC resampling procedure. As discussed in our response to Reviewer X5dr (*2. Estimator Variance and Convergence Behavior*), we compare the variance of different estimators under identical experimental settings and analyze their convergence behavior. This complementary experiment confirms that FP++ achieves more stable estimation and lower error, independent of SMC resampling effects. Therefore, while degeneracy is a known issue, FP++ directly mitigates it.
>
> **4. Scope of Applicability**
>
> We thank the reviewer for pointing out the need to clarify the practical scope of our method.
>
> Our approach requires both components at different stages:
> (i) samples from a target distribution (or dataset) to train the flow model, and
> (ii) an unnormalized energy function $u(x)$ to perform reweighting and recover the correct Boltzmann distribution.
>
> The most direct application is **molecular simulation**, where physical force fields naturally define $u(x)$ and yield Boltzmann distributions.
>
> We note that connections to reinforcement learning are primarily conceptual: while maximum-entropy formulations induce Boltzmann-form policies or trajectory distributions, they do not typically involve explicit Boltzmann sampling or flow-based modeling. We will revise the introduction to make these assumptions and scope more precise.
>
> **5. Notation and Presentation Improvements**
>
> We have improved clarity and consistency as follows:
>
> (1). Standardized notation: $b(x_t, t)$ for the drift, and $\mathbf{f}$ for the flow map;
>
> (2). Added missing references, including Rectified Flow and Stochastic Interpolants;
>
> (3). Clarified the bias of Hutchinson: while it provides an unbiased estimate of the log-density,the exponential reweighting introduces bias.

---

> > ### Author Rebuttal · Reviewer_AqpS · 2026-04-02
> >
> > I'd like to thank the reviewers for their response. I think the paper is a nice contribution for a relevant problem, and I maintain my score.

---

### Decision · Program_Chairs · 2026-04-30

**Decision:**

Reject

**Comment:**

This work introduces Flow Perturbation++, a method for estimating the Jacobian determinant of Continuous Normalizing Flow type models for Boltzmann sampling. Based on the recently introduced Flow Perturbation method, the new method injects noise and estimates the Jacobian during intermediate integration steps rather than just at the end. The method is evaluated on mode preservation on a 1000D Gaussian mixture model and the Chignolin protein system.

The reviewers initially raised concerns around computational cost, evaluation performance conflated with the SMC loop, and the use of a single probe in high dimensions. The authors addressed these concerns by performing additional experiments and clarifying various points, which by and large satisfied the reviewers. Two reviewers raised their scores but dropped their confidence. For a final scores of 4 4 4.

Given that no reviewer felt strongly about the work and I have extensive experience in recent Boltzmann sampling methods, I took a deeper look.
* Looking at the flow perturbation paper I was surprised to find that some figures are copied almost verbatim from the FP to FP++. Figure 2 of FP++ is the same as FP Figure 4 except panels A and B are swapped. While not central to the paper, to me this is extremely ethically borderline with unattributed recycling of figures. Similarly for Figure 5 of FP++ and Figure 7 of FP the plots are identical in some cases and almost identical in others.
* The novelty over FP is marginal at best in my view. Where FP takes the single step estimator and FP++ takes the multistep, the number of steps is not discussed at all. From personal experience the number of steps matters a great deal in the quality of likelihood estimates for Boltzmann sampling. Even with "exact" Jacobian calculation using autograd, an insufficient number of steps is still often enough to throw off the estimation.
* This work misses a substantial section of the more recent Boltzmann sampling work in the ML community.
  > Consequently, existing CNF-based methods have only achieved unbiased Boltzmann sampling for relative low-dimensional systems (Klein et al. 2023).

  misses work which scales CNF-based methods to Hexa-peptides and transferable across tetrapeptide systems (Tan et al. 2025ab).

* From the initial experiments it was impossible to tell how good the estimator given its conflation with SMC. The additional experiments run during the rebuttal show that with a single run FP++ has extremely large absolute error. I believe this would make it not useful for Boltzmann samplers relative to other SOTA methods. To me the error in finite difference step size (e.g. \~3 in 1000D with $\delta=0.001$) and (\~6 in 1000D for FP++) makes it uncompetitive with other approaches which are not benchmarked here. This error represents $e^3 \approx 20$ and $e^6 \approx 400$ error in the importance weights, which makes them fairly unusable.

* I'm also skeptical of the evaluation. This paper uses no established Boltzmann sampling benchmarks in the community and instead develops new ones without benchmarking other models. To me its unclear how difficult these scenarios are.

Because the algorithmic contribution is marginal, the absolute error remains prohibitively high for practical Boltzmann sampling, and there are severe concerns regarding the unacknowledged recycling of figures, at this time I recommend overriding the reviewers' weak scores and recommend rejection.